# Transient inhibition of the ERK pathway prevents cerebellar developmental defects and improves long-term motor functions in murine models of neurofibromatosis type 1

Edward Kim[1†], Yuan Wang[1,2,3†], Sun-Jung Kim[1,2], Miriam Bornhorst[1,2,4], Emmanuelle S Jecrois[1,2], Todd E Anthony[5], Chenran Wang[6], Yi E Li[1], Jun-Lin Guan[1,6], Geoffrey G Murphy[7,8], Yuan Zhu[1,2,3*]

[1]Division of Molecular Medicine and Genetics, Department of Internal Medicine, University of Michigan Medical School, Ann Arbor, United States; [2]Gilbert Family Neurofibromatosis Institute, Center for Cancer and Immunology Research and Center for Neuroscience Research, Children's National Medical Center, Washington, United States; [3]Department of Cell and Developmental Biology, University of Michigan Medical School, Ann Arbor, United States; [4]Division of Hematology and Oncology, Department of Pediatrics, Mott Children's Hospital, University of Michigan, Ann Arbor, United States; [5]Laboratory of Molecular Biology, Rockefeller University, New York, United States; [6]Department of Cancer Biology, University of Cincinnati College of Medicine, Cincinnati, United States; [7]Molecular and Behavioral Neuroscience Institute, University of Michigan Medical School, Ann Arbor, United States; [8]Department of Molecular and Integrative Physiology, University of Michigan Medical School, Ann Arbor, United States

*For correspondence: yzhu@childrensnational.org

†These authors contributed equally to this work

Competing interests: The authors declare that no competing interests exist.

**Abstract** Individuals with neurofibromatosis type 1 (NF1) frequently exhibit cognitive and motor impairments and characteristics of autism. The cerebellum plays a critical role in motor control, cognition, and social interaction, suggesting that cerebellar defects likely contribute to NF1-associated neurodevelopmental disorders. Here we show that *Nf1* inactivation during early, but not late stages of cerebellar development, disrupts neuronal lamination, which is partially caused by overproduction of glia and subsequent disruption of the Bergmann glia (BG) scaffold. Specific *Nf1* inactivation in glutamatergic neuronal precursors causes premature differentiation of granule cell (GC) precursors and ectopic production of unipolar brush cells (UBCs), indirectly disrupting neuronal migration. Transient MEK inhibition during a neonatal window prevents cerebellar developmental defects and improves long-term motor performance of *Nf1*-deficient mice. This study reveals essential roles of Nf1 in GC/UBC migration by generating correct numbers of glia and controlling GC/UBC fate-specification/differentiation, identifying a therapeutic prevention strategy for multiple NF1-associated developmental abnormalities.

## Introduction

Neurofibromatosis type 1 (NF1) is a genetically inherited disorder that afflicts 1 in 2700 newborns (*Evans et al., 2010*). NF1 is caused by loss-of-function mutations in the *NF1* tumor suppressor gene, which encodes neurofibromin, a negative regulator of proto-oncogene RAS (*Cichowski and Jacks, 2001*;

**eLife digest** Neurofibromatosis type 1 is a condition characterized by the growth of tumors along the nerves of the body. It is caused by mutations in a gene called *NF1*, which codes for a protein that normally works to inhibit the activity of another protein called Ras. In healthy cells, Ras is needed to stimulate the cells to grow and divide. However, if the Ras protein is not turned off at the right time or if it is activated at the wrong time, it can force cells to keep growing and dividing; this leads to the growth of tumors.

Along with being prone to developing cancer, individuals with neurofibromatosis type 1 also develop a range of neurodevelopmental disorders that alter their learning, motor skills and social interactions. Some also exhibit behaviors that are associated with autism. This led Kim, Wang et al. to investigate whether a region of the brain—called the cerebellum—that has recently been associated with autism is also affected in a mouse model of neurofibromatosis type 1.

The cerebellum is best known for its role in coordinating movement, although it also has functions in cognition, behavior and other processes. Ras is involved in the development of the cerebellum; and so Kim, Wang et al. asked whether the loss of the *Nf1* gene from cells in the mouse cerebellum might cause the neurodevelopmental defects associated with neurofibromatosis type 1.

*Loss of Nf1* during early (but not in late) development of the cerebellum disrupted the normal organization of the nerve cells (or neurons) into specific cell layers. These defects were caused, in part, by the over-growth of a type of supporting cell—called glia cells—at a specific developmental stage—that would normally form a scaffold to help neurons migrate to their correct position. Nf1 also controls the generation of the correct types of neurons in the right time and at right location during the early development of the cerebellum.

Next, Kim, Wang et al. treated newborn mice with a compound that inhibits Ras signaling via their mother's milk for 3 weeks. In mice with an inactive *Nf1* gene, the treatment helped to prevent some defects in the cerebellum and the mice had improved motor coordination several months later. Whether this could form the basis of a preventative treatment for neurodevelopmental disorders associated with neurofibromatosis type 1 in humans remains a question for future work.

---

*Upadhyaya and Cooper, 2012*). RAS mediates multiple signaling pathways including extracellular signal-regulated kinase (ERK) subfamily of mitogen-activated protein kinases (MAPK), phosphatidylinositol 3-kinase (PI3K) and mammalian target of rapamycin complex 1 (mTORC1) (*Schubbert et al., 2007*; *Mendoza et al., 2011*). In addition to the development of tumors in the peripheral and central nervous system (CNS), neurodevelopmental deficits are highly prevalent among children with NF1, negatively impacting cognition, motor function, and social interaction (*Hyman et al., 2005, 2006*; *Johnson et al., 2010*; *Krab et al., 2011*; *Lorenzo et al., 2011*; *Lehtonen et al., 2013*; *Walsh et al., 2013*; *Garg et al., 2013a, 2013b*; *Adviento et al., 2014*; *Champion et al., 2014*; *Plasschaert et al., 2014*). While cognitive impairments associated with NF1 have been well documented, motor dysfunction, social and behavioral deficits including autism spectrum disorders (ASD) have only recently been established as common features of NF1 in childhood (*Johnson et al., 2010*; *Krab et al., 2011*; *Lorenzo et al., 2011*; *Walsh et al., 2013*; *Garg et al., 2013a, 2013b*; *Champion et al., 2014*). Approximately 50–80% of children with NF1 have impairments in fine and gross motor function, which can be identified as early as at the toddler stage (*Johnson et al., 2010*; *Krab et al., 2011*; *Lorenzo et al., 2011*). One recent study has shown that impairments of gross motor skills and cognitive functioning in NF1 are often co-morbidities, suggesting the potential existence of a common pathological mechanism underlying both motor and cognitive impairments (*Champion et al., 2014*).

The cerebellum is traditionally known as a motor organ, which controls both motor coordination and motor learning (*Sillitoe and Joyner, 2007*). However, it has been increasingly recognized that the cerebellum also plays a critical role in higher-order brain functions such as cognition, learning, affect and behavior (*Schmahmann and Caplan, 2006*; *Strick et al., 2009*; *Basson and Wingate, 2013*). About 80% of individuals with ASD exhibit anatomical abnormalities in the cerebellum, among which gliosis and Purkinje cell loss are most commonly identified (*Bailey et al., 1998*; *Palmen et al., 2004*; *Vargas et al., 2005*; *Fatemi et al., 2012*). Magnetic resonance imaging (MRI) studies on children with ASD discovered increased white matter and reduced gray matter volumes in the cerebellum

(*Courchesne et al., 2001*; *Bloss and Courchesne, 2007*). In addition, T2-weighted hyperintensities in the cerebellum are one of the most consistent brain abnormalities observed in individuals with NF1 (*Feldmann et al., 2010*; *Payne et al., 2014*). These findings suggest that individuals with NF1, particularly those with co-morbidity of cognitive, motor and social deficits, might have developmental abnormalities of the cerebellum.

During development, major cerebellar cell populations are derived from two germinal zones. Radial glial (RG) stem cells in the ventricular zone of the fourth ventricle (IV–VZ) give rise to all the GABAergic neuronal lineages—Purkinje cells and GABAergic interneurons including small deep cerebellar nuclei (DCN), Golgi, basket, and stellate cells (*Sillitoe and Joyner, 2007*; *Buffo and Rossi, 2013*). Cerebellar astrocytes including Bergmann glia (BG) are also derived from RG cells in the IV-VZ. A secondary germinal zone in the anterior portion of the rhombic lip (RL) generates glutamatergic neuronal lineages, including large DCN, unipolar brush cells (UBCs), and granule cells (GCs) (*Englund et al., 2006*; *Sillitoe and Joyner, 2007*; *Mugnaini et al., 2011*; *Buffo and Rossi, 2013*). At the onset of mouse cerebellar neurogenesis, projection neurons (Purkinje cells and DCN) are first generated between embryonic day 10.5–11.5 (E10.5–11.5) (*Hatten, 1999*). At E12.5, Math1 (also known as Atoh1)-expressing proliferating precursors migrate out of the RL and spread across the surface of the cerebellum to form the external granule layer (EGL) (*Hatten, 1999*). Around postnatal day 4–5 (P4–5), GC precursors (GCPs) in the EGL start to exit cell cycle and form a pre-migratory zone along the edge of the EGL. Postmitotic cells then migrate along the BG scaffold, across the molecular layer (ML), and form the internal granule layer (IGL), where they become mature GCs and form synapses with Purkinje cells through mossy fibers (*Hatten, 1999*; *Sillitoe and Joyner, 2007*). Besides GCs, RL-derived Math1$^+$ precursors also generate the second type of glutamatergic excitatory interneurons, UBCs. UBCs exit the RL at perinatal stages and migrate through the white matter (WM) into the IGL of the cerebellum (*Englund et al., 2005*; *Mugnaini et al., 2011*). The molecular and cellular mechanisms regulating migration and lineage specification of GC/UBC precursors remain largely unknown.

It has been shown that the components of RAS/ERK and PI3K signaling pathways play diverse and critical roles in different cell types and at distinct stages during cerebellar development (*Yue et al., 2005*; *Fogarty et al., 2007*; *Lin et al., 2009*; *Yu et al., 2011*; *Müller Smith et al., 2012*; *Li et al., 2014*; *Meier et al., 2014*). However, recent studies have shown that *Nf1* heterozygous mice exhibit no defect in cerebellum-specific motor tests and that children with NF1 do not appear to have a general dysfunction of the cerebellum (*Krab et al., 2011*; *van der Vaart et al., 2011*). These observations suggest that mono-allelic *NF1/Nf1* inactivation is not sufficient to cause developmental and functional deficits in the cerebellum. Thus, we hypothesize that bi-allelic *Nf1* inactivation in neural stem and progenitor cells during development is required for producing cellular pathologies in the cerebellum. In this study, we utilize four genetically engineered mouse (GEM) models that target bi-allelic *Nf1* inactivation into different cell lineages or in the same lineage(s) at different stages during cerebellar development. We identify a specific time window of susceptibility of glial and neuronal precursors to *Nf1* mutation during early, but not late stages of cerebellar development. Accordingly, we develop a transient therapeutic strategy during a neonatal window, which prevents cerebellar developmental defects and provides long-term benefits on motor functions.

## Results

### Bi-allelic inactivation of *Nf1* at early developmental stages disrupts neuronal lamination in the cerebellum

We first utilized a Cre transgenic strain controlled by human glial fibrillary acidic protein promoter (hGFAP-cre) to inactivate *Nf1* in RG stem cells in the IV-VZ around E12.5 (*Figure 1—figure supplement 1A,A'*). RG cells targeted by hGFAP-cre in the E12.5 IV-VZ give rise to glutamatergic neuronal precursors populating the RL and astrocyte precursors producing astrocytes in cerebellar parenchyma (*Zhu et al., 2005*; *Sillitoe and Joyner, 2007*; *Yang et al., 2008*). Of note, the majority of oligodendrocytes in the cerebellum do not arise from RG cells in the IV-VZ, but from sources in the ventral-lateral mid/hindbrain (that was also targeted by hGFAP-cre) (*Yang et al., 2008*; *Grimaldi et al., 2009*; *Mecklenburg et al., 2011*). Glutamatergic neurons (including GCs) and glia, but not GABAergic Purkinje cells, were targeted by hGFAP-cre-mediated recombination, revealed by β-galactosidase (β-gal) expression of the Rosa26-LacZ reporter and PCR analysis (*Figure 1—figure supplement 1A,B,B'*) (*Soriano, 1999*). The hGFAP-cre driven *Nf1* conditional knockout mice

(hereafter referred to as *Nf1*hGFAPCKO) displayed severe defects in motor behaviors including abnormal hind limb crossing and vertical projection of tails. Most of the *Nf1*hGFAPCKO mice had ataxic hind limbs and some exhibited a unique 'handstand' behavior during postnatal stages (*Figure 1—figure supplement 1C*; *Video 1*). The motor dysfunction observed in neural-specific *Nf1*hGFAPCKO mice suggests possible defects in the cerebellum. As revealed by Hematoxylin & Eosin (H&E) staining, neuronal lamination in all folia of the *Nf1*hGFAPCKO cerebellum was disrupted with a varying degree of severity (*Figure 1A*). In the severely affected areas (e.g., folia V/VI), the patterning of Calbindin+ Purkinje cells was disrupted, and failed to form the single Purkinje cell layer (PCL) observed in controls (*Figure 1B*). Since hGFAP-cre was not targeted to Purkinje cells, the disorganized Purkinje cells still expressed neurofibromin (*Figure 1C*; *Figure 1—figure supplement 1B*) (*Zhu et al., 2005*; *Yang et al., 2008*). Thus, the disrupted patterning of Purkinje cells must be non-cell-autonomous, and secondary to *Nf1* inactivation in other cerebellar neurons and/or glia. To further confirm the notion, we utilized L7-cre to inactivate *Nf1* specifically in Purkinje cells (*Figure 1—figure supplement 1D*) (*Tsai et al., 2012*). No overt defect was identified in the *Nf1*L7CKO cerebellum, confirming that Nf1 is relatively dispensable in the development of Purkinje cells (*Figure 1—figure supplement 1E,F*).

In contrast to the normal cerebellum with three distinct laminar layers—ML, PCL and IGL, the boundary between these three layers in the *Nf1*hGFAPCKO cerebellum was less clearly defined, and the cell number in the ML was dramatically increased with a concomitant reduction of cells in the IGL (*Figure 1A–A''*; *Figure 1—figure supplement 1A'*). Some of the abnormally accumulated cells in the *Nf1*hGFAPCKO ML were found on the pial surface of the cerebellum, corresponding to the EGL—a structure that transiently exists during development, but disappears after the completion of GC migration (arrows, *Figure 1Ab''*). No significant difference was found in the total number of NeuN+ neurons in the ML and IGL (per surface area) between the control and *Nf1*hGFAPCKO cerebellum. Instead, the distribution of NeuN+ neurons was greatly altered between the control and *Nf1*hGFAPCKO cerebellar cortex—the number of NeuN+ neurons was increased over 20-fold in the mutant ML (*Figure 1D,D'*). While no cells in the control ML expressed GABA receptor alpha 6 subunit (GABARα6), a marker for mature GCs, 60% of abnormally accumulated neurons in the *Nf1*hGFAPCKO ML were positive for GABARα6 (*Figure 1D,D''*). These observations are most consistent with a model wherein a subset of *Nf1*-deficient GCs fail to migrate inward from the EGL, across ML and PCL, to the IGL. The arrested migration of GCs leads to abnormal neuronal patterning and lamination of the *Nf1*hGFAPCKO cerebellum.

We next investigated the integrity of the BG scaffold—an essential glial structure for GC migration during development (*Sillitoe and Joyner, 2007*). In the control cerebellum, BG cells were located either immediately below or interspersed between Purkinje cell bodies, forming a Purkinje cell and BG (PC/BG) monolayer (arrows, *Figure 1E,F*) (*Buffo and Rossi, 2013*). In contrast, *Nf1*-deficient BG cells failed to form a PC/BG monolayer, which appeared to be caused by excess number and mispositioning of BG cells (arrows, *Figure 1E,E',F*). A strong correlation was observed between the severity of neuronal patterning defects and the abnormalities in the number and position of BG cells and the integrity of the PC/BG monolayer (*Figure 1F*). Importantly, the capacity of *Nf1*-deficient BG cells to extend their processes to the pial surface was relatively uncompromised—this was even true for some of the BG cells that were located in the areas outside the PC/BG layer in the *Nf1*hGFAPCKO cerebellum (*Figure 1G*). Based on these results, we propose a model wherein inactivation of *Nf1* during early embryonic stages overproduces BG

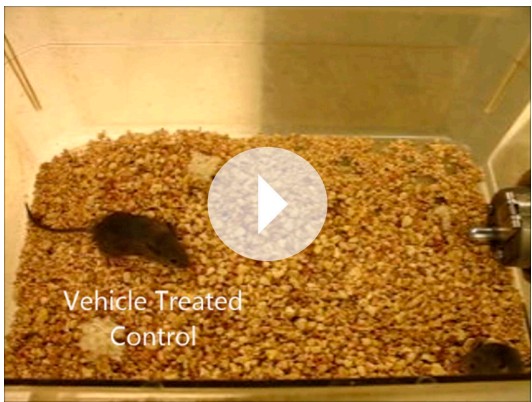

**Video 1**. Behavioral defects of *Nf1*hGFAPCKO mice before and after MEKi treatment. This video illustrates the typical phenotypes seen in vehicle-treated and untreated *Nf1*hGFAPCKO mice compared to controls including scruffy fur, unbalanced gait, vertical projection of the tail, hyperactivity, hind-limb crossing, and hand-standing. Following MEKi treatment, the G-responders appear similar to the controls except the vertical projection of the tail, while the P-responders are more similar to the vehicle-treated mutants. All the mice in this video are P21 except the mouse with the hand-standing phenotype, which was videotaped at 4 months of age.

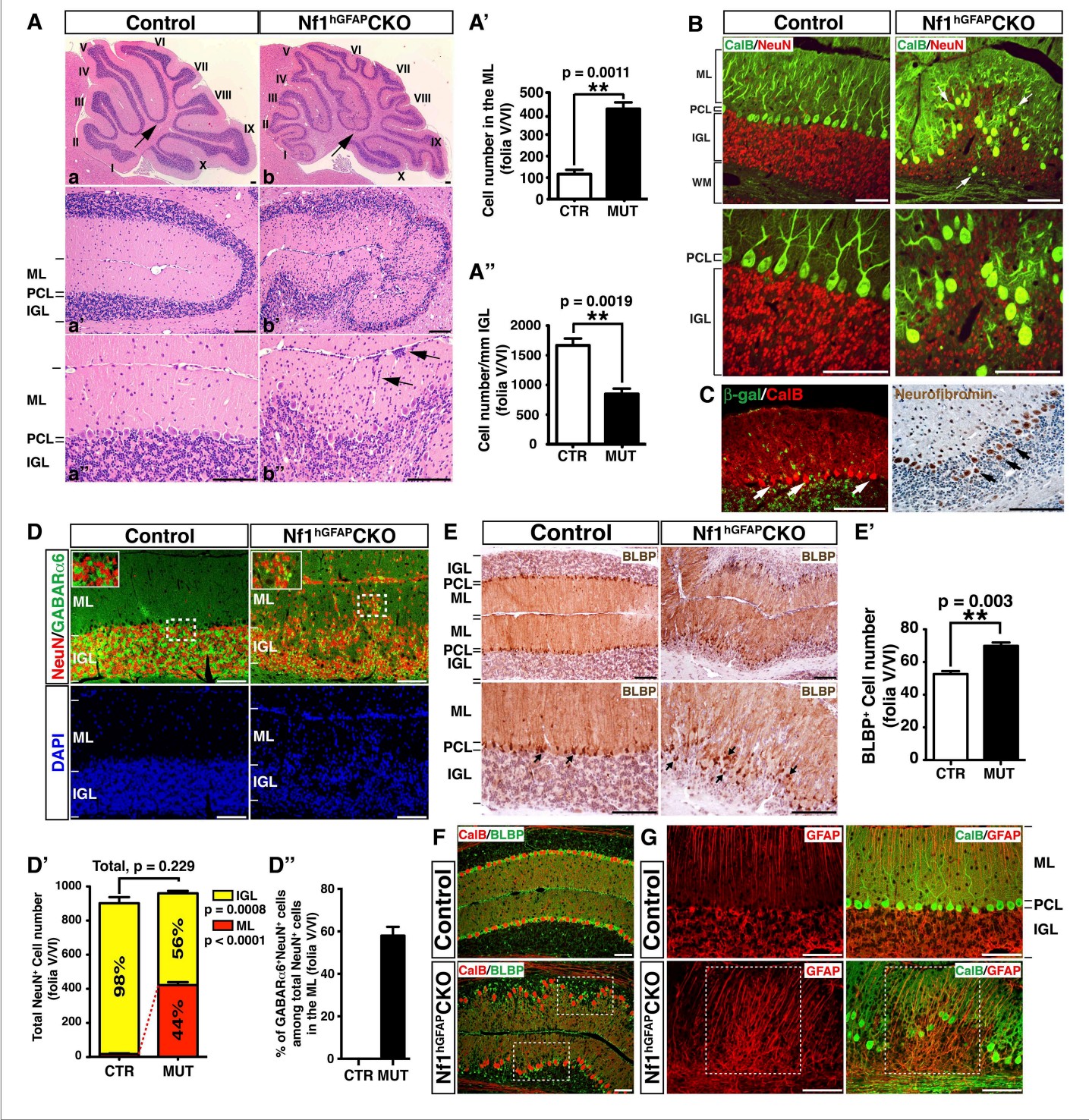

**Figure 1**. *Nf1* inactivation during early cerebellar development disrupts neuronal lamination and causes BG abnormalities. (**A**) Sagittal sections from control and *Nf1*hGFAPCKO cerebella at 2 months of age were stained with H&E and imaged at three increasing magnifications. Each cerebellar folium is numbered by Roman numerals. Arrows in (**A**, **B**) highlight folia V/VI where mutants exhibited extra foliation and arrows in (**B''**) point to the cells clustered in the ML and on the pial surface of the *Nf1*hGFAPCKO cerebellum. The total number of cells in the ML and IGL were quantified in (**A'**) and (**A''**). (**B**) Cerebellar sections were stained for Calbindin (CalB) /NeuN. Arrows point to mispositioned CalB+ Purkinje cells. (**C**) Sections from adult *Nf1*hGFAPCKO cerebellum were stained with β-gal/CalB and Nf1. Purkinje cells (arrows) were β-gal− and expressed Nf1. (**D**) Cerebellar sections were stained for NeuN and GABARα6. The insets are the high magnification images of the boxed areas highlighting the co-localizing cells. The number of NeuN+ cells in the

*Figure 1. Continued on next page*

*Figure 1. Continued*

ML and IGL were quantified in (**D'**) and the percentage of NeuN$^+$ cells in the IGL among total NeuN$^+$ cells (ML + IGL) is shown. The percentage of GABAR$\alpha$6$^+$NeuN$^+$/NeuN$^+$ cells in the ML was quantified in (**D''**). (**E**) The cell bodies (arrows) and processes of BG in control and mutant cerebella were labeled by BLBP staining. The total number of BLBP$^+$ BG cells was quantified in (**E'**). (**F**) Sections were stained for CalB/BLBP. Boxed areas compare the Purkinje cell alignment in the less affected areas (lower box) and severely affected areas (upper box) in the *Nf1*$^{hGFAP}$CKO cerebellum. Note the correlation between increased number/misalignment of BG and the severity of the disruption of Purkinje cell layer. (**G**) Sections were stained for CalB/GFAP. Boxed areas highlight the severely disrupted BG alignment and Purkinje cell patterning. All the quantification data are presented as mean ± SEM. DAPI labels the nuclei. Scale bars: 50 µm.

The following figure supplement is available for figure 1:

**Figure supplement 1**. Purkinje cell defects in the *Nf1*$^{hGFAP}$CKO cerebellum are non-cell-autonomous.

cells, thus impairing the formation of the PC/BG monolayer, and consequently, the alignment of the BG scaffold. The defective BG scaffold non-cell-autonomously arrests GC migration and subsequently disrupts Purkinje cell patterning during cerebellar development.

## *Nf1* inactivation increases astrocyte proliferation at an intermediate precursor stage

We investigated the mechanism by which *Nf1* inactivation during early cerebellar development over-produced BG cells in the *Nf1*$^{hGFAP}$CKO cerebellum. No defect was observed in the overall structure and the distribution of RL-derived Math1$^+$ neuronal precursors in the EGL, Purkinje cells or BLBP$^+$ cells in the E17.5 *Nf1*$^{hGFAP}$CKO cerebellum (*Figure 2—figure supplement 1A*). Furthermore, no difference was identified in the number of BLBP$^+$ RG cells (including proliferating BLBP$^+$Ki67$^+$ population) in the IV-VZ (*Figure 2—figure supplement 1B*). However, an increased number of both total BLBP$^+$ and proliferating BLBP$^+$Ki67$^+$ cells was observed in the prospective white matter (PWM) of the E17.5 *Nf1*$^{hGFAP}$CKO cerebellum compared to controls (*Figure 2A,A'*). More importantly, mitotic index, determined by the ratio of proliferating BLBP$^+$Ki67$^+$ cells to the total BLBP$^+$ cells, was significantly increased in the E17.5 *Nf1*$^{hGFAP}$CKO PWM (*Figure 2A,A'*). These observations suggest that Nf1 specifically constrains the proliferation of astrocyte precursors in the PWM, but not multipotent RG cells of the IV-VZ in the E17.5 cerebellum. At P0.5, both Purkinje cells and BLBP$^+$ BG cells (except for those populating future folia IX and X) were situated into a more defined Purkinje cell plate (or rudimentary PC/BG layer); BG cells, particularly those in the anterior cerebellum, underwent differentiation extending their GFAP$^+$BLBP$^+$ fibers to the pial surface (*Figure 2—figure supplement 1C*). Unexpectedly, the increased mitotic index was no longer detected in the more differentiated BLBP$^+$ BG cells in the anterior folia, but only in the posterior folia (IX and X) of the *Nf1*$^{hGFAP}$CKO cerebellum where BG cells had not formed a rudimentary PC/BG layer at P0.5 (*Figure 2B,B',2C,C'*; *Figure 2—figure supplement 1D*). The differentiation of BG cells was accompanied by the transition of (1) the anatomical location from the PWM to rudimentary PC/BG layer and (2) the subcellular expression of BLBP from nuclei to cytoplasm. Specifically, BLBP staining in the anterior folia of the P0.5 control cerebellum became largely cytoplasmic, while a robust nuclear BLBP staining was typically found in less differentiated BLBP$^+$ cells in both the E17.5 control and *Nf1*$^{hGFAP}$CKO cerebellum (arrows, *Figure 2A*). Consistently, an increase of cells with strong nuclear BLBP staining in the P0.5 *Nf1*$^{hGFAP}$CKO cerebellum was identified, suggesting a persistent increase of proliferating BG cells into postnatal stages (*Figure 2D*). Similarly, BLBP$^+$ cells in the PWM, the precursors of WM astrocytes, still exhibited an increased mitotic index in the P0.5 *Nf1*$^{hGFAP}$CKO cerebellum (*Figure 2—figure supplement 1D–D''*). At P8 when BG cells had formed a rudimentary PC/BG layer throughout the cerebellum, the mitotic index of BLBP$^+$ BG cells was no longer increased in the *Nf1*$^{hGFAP}$CKO cerebellum, despite a significant increase of both total BLBP$^+$ and proliferating BLBP$^+$Ki67$^+$ BG cells (*Figure 2—figure supplement 1E,E'*). Together, these results suggest that Nf1 constrains the proliferation of the BG/astrocyte lineage cells specifically at an intermediate precursor stage located in the PWM, but not at an earlier RG stem cell stage or a more differentiated BG stage, thereby identifying a window of susceptibility of astrocyte precursors to *Nf1* mutation.

## Nf1 is not required for late stages of BG development

At P8, control BG cells labeled by nuclear Cre staining formed a rudimentary 2 to 3 cell-layer lining within the APC$^+$ Purkinje cell layer (*Figure 2E*). In contrast, excess BG cells failed to form a similar

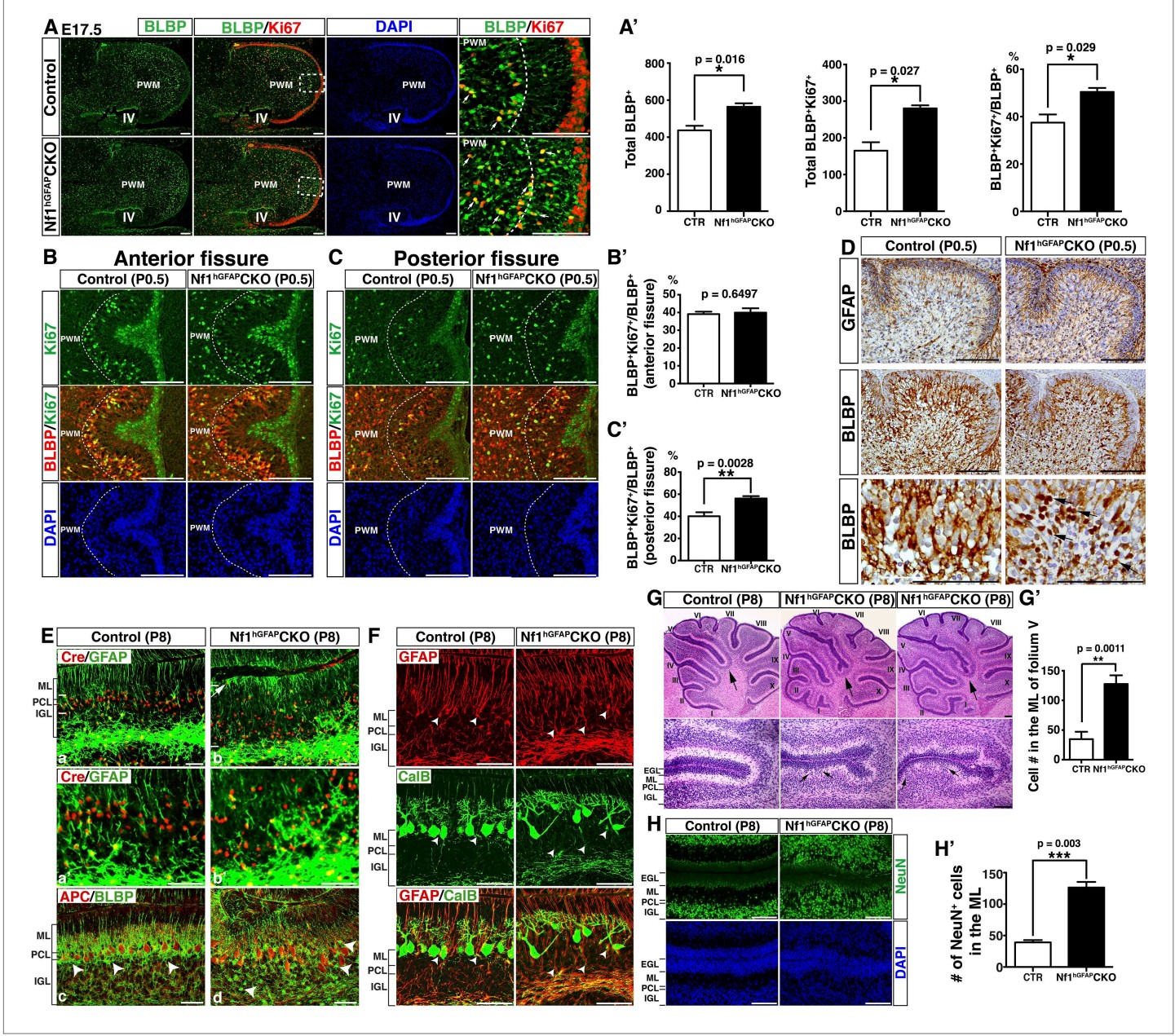

**Figure 2**. *Nf1* inactivation leads to increased proliferation of intermediate glial precursors and subsequent glial/neuronal defects during perinatal development. (**A**) Sections of control and *Nf1*[hGFAP]CKO cerebella were stained for BLBP and Ki67. Boxed areas in (**A**) are shown in high magnification on the right. Dashed lines delineate the boundary between the prospective white matter (PWM) and cerebellar cortex. IV, fourth ventricle. The number and mitotic index of BLBP+ cells were quantified in (**A'**). (**B, C**) Cerebellar sections were stained for BLBP/Ki67 and imaged at anterior fissure (**B**) and posterior fissure (**C**). Dashed lines delineate the boundary of the PWM. The mitotic index of BLBP+ cells in respective fissures was quantified in (**B'**) and (**C'**). (**D**) Sections were stained for GFAP and BLBP. BLBP in the mutant cerebellum exhibit strong nuclear staining (arrows). (**E**) Sections were stained for Cre/GFAP (**A**–**B'**) and APC/BLBP (**C, D**). White bars mark the rudimentary PC/BGL in the control and the *Nf1*[hGFAP]CKO cerebellum. The arrow points to an abnormal BG cell located close to the EGL in the mutant cerebellum. (**C, D**) Arrowheads point to BG cell bodies. (**F**) Sections were stained for CalB/GFAP. Arrowheads point to BG cell bodies. (**G**) Sections were stained by H&E. Lower panels are the high magnification view of folia V/VI. Arrows point to folia V/VI (upper panels) where the mutants exhibit the extra foliation. In lower panels, arrows point to the cell clusters in the mutant ML. The total cell number in the ML was quantified in (**G'**). (**H**) Sections were stained for NeuN. The total number of NeuN+ cells in the ML was quantified in (**H'**). All the quantification data are presented as mean ± SEM. DAPI labels the nuclei. Scale bars: 50 μm.

The following figure supplement is available for figure 2:

**Figure supplement 1**. Nf1 regulates cerebellar astrocyte proliferation during an immediate precursor stage.

PC/BG layer, but were aberrantly distributed throughout the $Nf1^{hGFAP}$CKO cerebellar cortex (*Figure 2E,F*). Consequently, some *Nf1*-deficient BG cells extended their processes from ectopic locations of the cerebellum including the WM, ML and IGL, accompanied by misalignment of the BG scaffold (*Figure 2E,F*; *Figure 2—figure supplement 1F*). Similar to the $Nf1^{hGFAP}$CKO cerebellum in adulthood, disrupted patterning of GCs and Purkinje cells often co-existed with the misalignment of BG cells at P8 (*Figure 2F*; *Figure 2—figure supplement 1F*). Indeed, all major defects observed in BG cells, GCs and Purkinje cells of the adult $Nf1^{hGFAP}$CKO cerebellum were identified at P8, suggesting that these defects manifest during neonatal stages (*Figure 2G–H'*; *Figure 2—figure supplement 1F–H''*). These results supports a model wherein overproduction of astrocyte precursors during embryonic stages is the primary defect causing both glial and neuronal abnormalities in the $Nf1^{hGFAP}$CKO cerebellum. To rule out a role of Nf1 in late stages of glial development, we sought to more specifically inactivate *Nf1* in glial cell lineages during late embryonic stages by employing an inducible system. This inducible system utilized a Nestin-cre$^{ER}$ (NCreER) transgene, which can be activated in Nestin-expressing neural stem and progenitor cells by tamoxifen (TM) treatment (*Figure 3—figure supplement 1A*) (*Burns et al., 2007*; *Wang et al., 2012*). Upon a single TM pulse at E17.5, NCreER-targeted cells analyzed at P21 were restricted to glial lineage cells in both the control and $Nf1^{NCreER}$CKO cerebellum with only one exception in folium X, where a significant number of both glia and GCs in the IGL were targeted (*Figure 3A,B*; *Figure 3—figure supplement 1B,C*). Given the sequential differentiation from the anterior to posterior cerebellum, this targeting strategy would inactivate *Nf1* in more differentiated glial cells in the anterior folia compared to those in the posterior folia. No alteration in the number of BLBP$^+$ cells or the BG scaffold was observed in any of the anterior folia (I to V) between the control and $Nf1^{NCreER}$CKO cerebellum (*Figure 3A*). Furthermore, neurons did not abnormally accumulate in the ML of the anterior $Nf1^{NCreER}$CKO folia (*Figure 3A,A'*; *Figure 3—figure supplement 1D,D'*). In folium X, the number of *Nf1*-deficient BLBP$^+$ cells (BLBP$^+$β-gal$^+$) was significantly increased, though this was subtle compared to the $Nf1^{hGFAP}$CKO cerebellum (*Figure 3B,C*). Indeed, the other posterior folia (e.g., VI–IX) exhibited a slightly, but significantly increased number of *Nf1*-deficient BLBP$^+$ cells (*Figure 3—figure supplement 1C,C'*). The overproduction of BLBP$^+$β-gal$^+$ BG cells in the posterior folia of the $Nf1^{NCreER}$CKO cerebellum appeared to be caused by cell-autonomous *Nf1* inactivation, as the number of non-NCreER-targeted BLBP$^+$β-gal$^-$ cells in the same areas was not altered (*Figure 3B,C*). Importantly, the overproduction of BLBP$^+$ BG cells was accompanied by focal misalignment of the BG scaffold and abnormal accumulation of both *Nf1*-deficient (NeuN$^+$β-gal$^+$) and *Nf1*-intact (NeuN$^+$β-gal$^-$) neurons in the ML of the $Nf1^{NCreER}$CKO cerebellum (*Figure 3B,D*; *Figure 3—figure supplement 1E*). These results suggest that overproduction of this particular astrocytic lineage causes impaired formation of the PC/BG monolayer and BG scaffold, non-cell-autonomously arresting GC migration.

## Cell-autonomous function of Nf1 in the glutamatergic interneuronal lineages

To determine whether Nf1 also plays a cell-autonomous role in the glutamatergic neuronal lineages, we utilized Math1-cre, which specifically targets Math1$^+$ neuronal precursors in the developing RL that give rise to both GCs and UBCs (*Englund et al., 2006*; *Schuller et al., 2008*). Math1-cre-mediated recombination was almost exclusively restricted to the cells in the IGL of the adult cerebellum (*Figure 4A*). Double staining of β-gal and other lineage markers showed that β-gal$^+$ cells were restricted to GCs in the IGL, but not Purkinje cells or glial cells, including BG cells (*Figure 4B,B'*). No defect in Purkinje cells or glial cells, particularly the BG scaffold, was observed in the adult $Nf1^{Math1}$CKO cerebellum (*Figure 4B,B'*). Consequently, unlike the severely disrupted $Nf1^{hGFAP}$CKO cerebellum, the $Nf1^{Math1}$CKO cerebellum was relatively unimpaired (*Figure 4—figure supplement 1A*). However, the $Nf1^{Math1}$CKO cerebellum did exhibit one major defect observed in the $Nf1^{hGFAP}$CKO cerebellum—abnormal accumulation of cells in the ML including some on the pial surface (arrows, *Figure 4A,B–B''*; *Figure 4—figure supplement 1A,A'*). The number of NeuN$^+$β-gal$^+$, but not NeuN$^+$β-gal$^-$ neurons, was significantly increased, suggesting that the increased neuronal density in the ML of the $Nf1^{Math1}$CKO cerebellum was caused by abnormal accumulation of *Nf1*-deficient NeuN$^+$ neurons in a cell-autonomous manner (*Figure 4A,B,B''*). Despite that no defect in the BG scaffold was observed, abnormal accumulation of neurons was evident in the ML of the P8 $Nf1^{Math1}$CKO cerebellum, suggesting a developmental defect during neonatal stages (*Figure 4—figure supplement 1B*). To investigate the mechanism underlying the arrested migration of *Nf1*-deficient GCs, we performed BrdU pulse-chase experiments to determine the time course of the differentiation and migration of newly generated GCs during neonatal

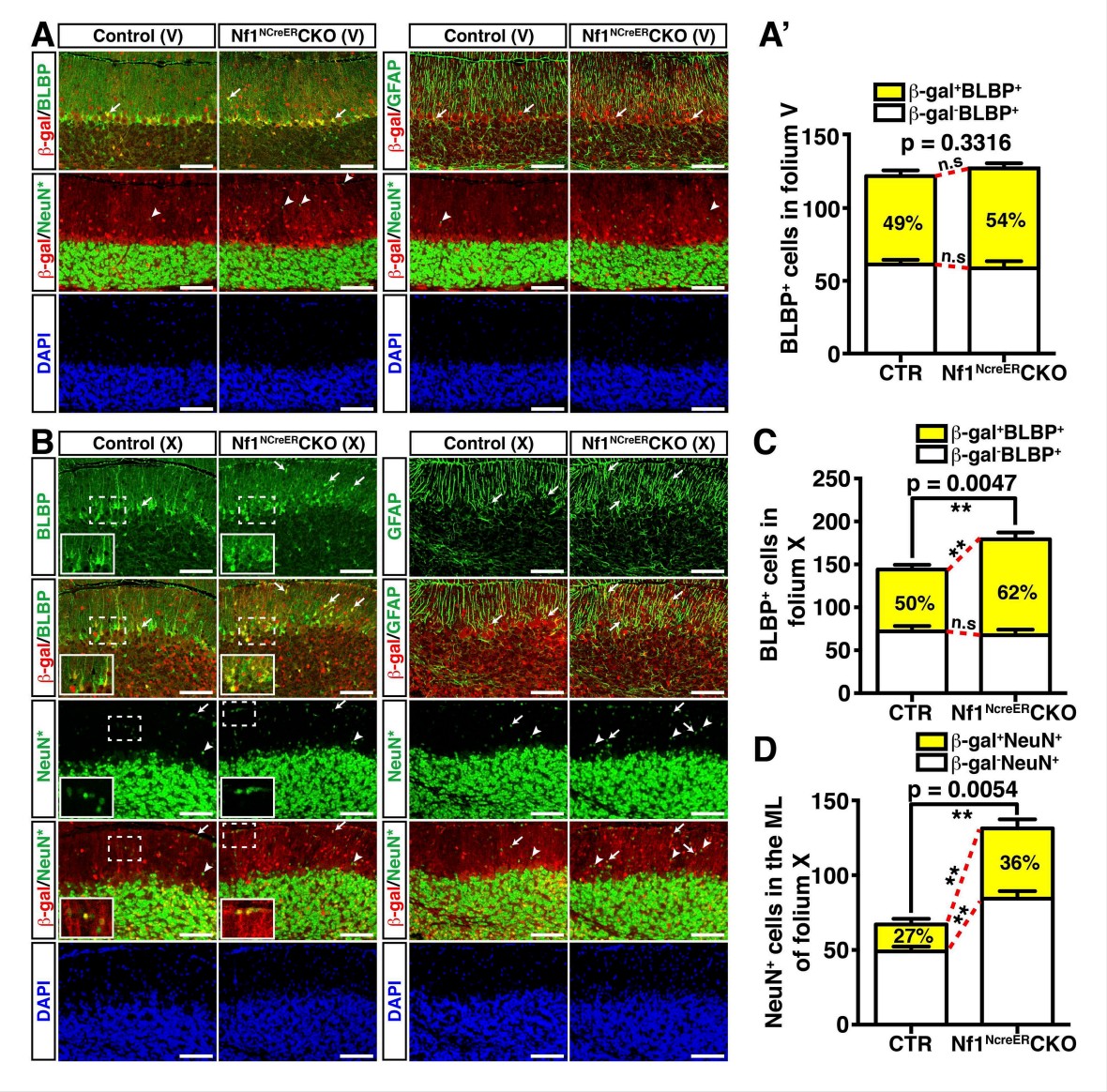

**Figure 3**. Nf1 is not required for BG function in late developmental stages. Control and *Nf1*[NcreER]CKO mice were tamoxifen (TM)-induced at E17.5 and analyzed at P21. Adjacent sections were triple-stained for BLBP/β-gal/NeuN or GFAP/β-gal/NeuN staining, and were imaged at folium V (**A**) and folium X (**B**). NeuN staining (*, Alexa 647) was artificially converted to green. The insets in (**B**) highlight BLBP⁺β-gal⁺ cells and NeuN⁺β-gal⁺ cells in the boxed areas. The number of BLBP⁺ cells (**A'**, **C**) and NeuN⁺ cells (**D**) was quantified and their co-localization with β-gal was compared. Note that in folia V, none of the NeuN⁺ cells co-localized with β-gal. Arrows point to representative co-localizing cells, and arrowheads label non-co-localizing cells. All the quantification data are presented as mean ± SEM. DAPI labels the nuclei. Scale bars: 50 μm.

The following figure supplement is available for figure 3:

**Figure supplement 1**. Inactivation of *Nf1* in Nestin-expressing progenitor cells at E17.5 induces overproduction of glia cells in the posterior cerebellar folia.

stages. We analyzed P8 control and *Nf1*[Math1]CKO cerebella 24, 32 and 48 hr after BrdU pulse by determining the location and differentiation status of BrdU-labeled cells. 24 hr after pulse, BrdU⁺ cells in both the control and *Nf1*[Math1]CKO cerebellum remained exclusively in the EGL and exhibited similar cell cycle profiles, suggesting that Nf1 is not required for proliferation and cell-cycle exit of GC precursors (GCPs) (*Figure 4—figure supplement 1C–C″*). However, 32 hr after BrdU pulse, a subset of BrdU⁺ cells in the control cerebellum started departing from the EGL and entered the ML, not yet reaching the IGL (*Figure 4C*). While migrating GCs in the control ML did not express MAP2, a marker for mature

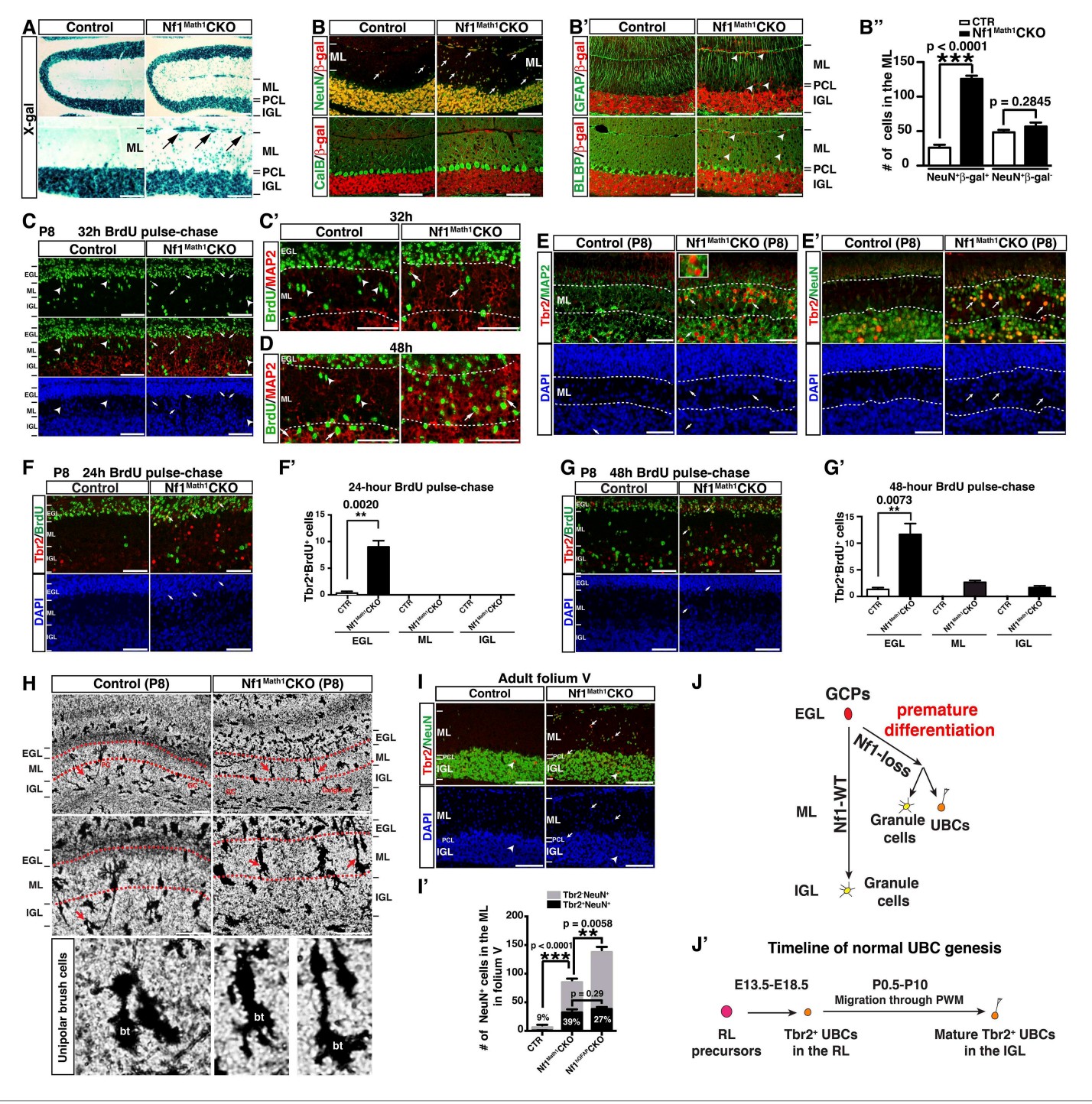

**Figure 4**. Neuron-specific *Nf1* inactivation leads to cell-autonomous defects in the glutamatergic interneuron lineages. (**A**) X-gal staining on P21 control and *Nf1*[Math1]CKO cerebella labeled cells that underwent Math1-cre-mediated recombination. Arrows point to the abnormally accumulated cells in the ML of the mutant cerebellum. (**B**, **B'**) Cerebellar sections were stained for NeuN/β-gal and Cal/β-gal, and GFAP/β-gal and BLBP/β-gal. The total number of NeuN[+]/β-gal[+] and NeuN[+]/β-gal[−] cell per high magnification field was quantified in (**B''**). (**C–D**) Control and *Nf1*[Math1]CKO mice were analyzed 32 hr and 48 hr after BrdU-pulse at P8. Cerebellar sections were stained for BrdU/MAP2 and imaged at folium V. High magnification images comparing the nuclear morphology of BrdU[+] cells are shown in (**C'**) and (**D'**). (**E**, **E'**) Cerebella sections from P8 control and *Nf1*[Math1]CKO mice were stained for Tbr2/MAP2 (**E**) and Tbr2/NeuN (**E'**) and imaged at folium V. The inset in (**E**) shows an example of Tbr2[+]MAP2[+] cells in the mutant ML. (**F**, **G**) 24 hr and 48 hr after BrdU-pulse, cerebellar sections from P8 control and *Nf1*[Math1]CKO mice were stained for Tbr2/BrdU. The distribution of Tbr2[+]BrdU[+] cells (arrows) in the EGL-ML-IGL

*Figure 4. Continued on next page*

Figure 4. Continued

was quantified 24 hr and 48 hr after BrdU-pulse in (**F′**) and (**G′**), respectively. (**H**) Golgi staining was performed on P8 cerebellar sections. Cells with UBC morphology are highlighted by red arrows and shown in high magnification (lower panels). Dashed red lines delineate the ML. PC, Purkinje cell; GC, granule cell; bt, the brush tip of UBCs. (**I**) Cerebellar sections from adult control and *Nf1*[Math1]CKO cerebella were stained for Tbr2/NeuN and imaged at folium V. (**I′**) The number of Tbr2+NeuN+ (arrows) and Tbr2+NeuN− cells (arrowheads) in the ML of folium V per high magnification image was quantified and compared with *Nf1*[hGFAP]CKO mice. The percentage of Tbr2+NeuN+ cells among total NeuN+ cells was also quantified. (**J**) A proposed model summarizes the role of Nf1 in preventing the ectopic differentiation of GCPs into UBCs. As a comparison, the timeline of normal UBC genesis is shown in (**J′**). All the quantification data are presented as mean ± SEM. DAPI labels the nuclei. Scale bars: 50 μm.

The following figure supplement is available for figure 4:

**Figure supplement 1**. Neuron-specific *Nf1* inactivation does not impact on the proliferation or migration of GCPs.

GCs in the IGL, and had nuclei with a spindle-shaped migrating morphology, many BrdU+ cells in the *Nf1*[Math1]CKO ML already acquired MAP2 expression with rounded nuclei—typical neuronal differentiation features for mature GCs in the IGL (arrows, *Figure 4C,C′*). After 48-hr BrdU pulse, BrdU+MAP2+ cells in the *Nf1*[Math1]CKO cerebellum progressively accumulated in the ML and EGL whereas control BrdU+ cells did not express MAP2 until they reached the IGL (*Figure 4D*; *Figure 4—figure supplement 1D,D′*). Newly generated neurons were abnormally accumulated in the ML/EGL of the *Nf1*[Math1]CKO cerebellum, leading to an increased number of differentiated BrdU+Ki67− and p27+ cells in the ML/EGL with a concomitant reduction in the IGL (*Figure 4—figure supplement 1E,E′,F,F′*). These observations suggest that a subset of *Nf1*-deficient GCPs undergo accelerated and premature neuronal differentiation, thereby arresting their migration in the EGL and ML. This conclusion was further supported by in vitro studies. 2 days after being plated in culture, significantly more *Nf1*-deficient GCPs expressed MAP2 than control cells, confirming an accelerated neuronal differentiation rate in a cell-autonomous manner (*Figure 4—figure supplement 1G,G′*). Importantly, *Nf1*-deficient GCs could efficiently interact with wild-type glia in a co-culture system and migrated normally in a trans-well migration assay (*Figure 4—figure supplement 1H,H′*). Together, both in vivo and in vitro assays argue that instead of causing defects directly on migration, *Nf1* inactivation promotes premature neuronal differentiation of a subset of GCPs, leading to abnormal accumulation of differentiated neurons in the ML and EGL.

## Fate-specification defects in *Nf1*-deficient glutamatergic neuronal lineages

Strikingly, many of the prematurely differentiated neurons in the ML of the P8 *Nf1*[Math1]CKO cerebellum expressed Tbr2, a marker specific for UBCs in the cerebellum (*Figure 4E,E′*). UBCs are the other known RL-derived glutamatergic interneurons that are enriched in folia responsible for vestibular motor function (folia IX and X) (*Mugnaini et al., 2011*). While GCs arise from GCPs in the EGL and migrate across ML to IGL between P4–P21, Tbr2+ UBCs exit the RL during late embryonic stages, and then migrate through the PWM into the IGL until P10 (*Englund et al., 2006*; *Sillitoe and Joyner, 2007*; *Mugnaini et al., 2011*). While Tbr2+ UBCs were rarely identified in the control ML (*Englund et al., 2006*), Tbr2+ cells including the newly generated Tbr2+BrdU+ cells were readily identified in the P8 *Nf1*[Math1]CKO EGL and ML (*Figure 4E,E′,F,F′*). Tbr2 expression was ectopically acquired in a subset of newly generated cells even before migrating out of the EGL, and over time these Tbr2+BrdU+ double-positive cells progressively accumulated from the EGL, ML to IGL (*Figure 4F,F′,G,G′*). These observations demonstrate that *Nf1* inactivation ectopically generates Tbr2+ UBCs from GCPs in the EGL, which appear to migrate along the same pathway as GCs. More importantly, cells with the classic morphology of UBCs were identified in the ML of the P8 *Nf1*[Math1]CKO cerebellum, but not in the control cerebellum, indicating that these Tbr2+ cells not only acquire UBC-specific Tbr2 expression, but also undergo morphological differentiation of UBCs (*Figure 4H*) (*Mugnaini and Floris, 1994*). Consequently, approximately 30–40% of abnormally accumulated NeuN+ neurons in the ML of the adult *Nf1*[Math1]CKO cerebellum expressed Tbr2 (*Figure 4I,I′*). Together, these observations identify cell-autonomous Nf1 functions in suppressing GCPs to ectopically differentiate into UBCs (*Figure 4J,J′*).

## Ectopic generation of Tbr2+NeuN+ UBCs is responsible for non-GC accumulation in the ML of the adult *Nf1*[hGFAP]CKO cerebellum

Ectopic generation of Tbr2+ UBCs from *Nf1*-deficient GCPs prompted us to investigate whether GABARα6-negative neurons abnormally accumulated in the ML of the *Nf1*[hGFAP]CKO cerebellum were

UBCs (*Figure 1D,D''*). Indeed, a dramatic increase of Tbr2$^+$ cells was also observed in the ML throughout the adult *Nf1*$^{hGFAP}$CKO cerebellum, which was most evident in folia V/VI (*Figure 5A–A''*). The increase of Tbr2$^+$ cells in the ML (98%) and IGL (94%) of the *Nf1*$^{hGFAP}$CKO cerebellum was almost entirely contributed by UBCs that expressed a moderate level of Tbr2 as well as NeuN (Tbr2$^+$NeuN$^+$) (arrows, *Figure 5B,B'*). Almost all UBCs in the control folium V expressed high levels of Tbr2, but no NeuN (Tbr2$^{high}$NeuN$^-$) and were mainly distributed in the IGL (arrowheads, *Figure 5B,B'*). It should be noted that the number of Tbr2$^{high}$NeuN$^-$ UBCs was not significantly different between the control and the *Nf1*$^{hGFAP}$CKO cerebellum. The distinction between Tbr2$^+$NeuN$^+$ and Tbr2$^{high}$NeuN$^-$ UBC populations was further supported by the different percentage of these two types of UBCs expressing a high level of Calretinin (CR$^{high}$). CR$^{high}$ is a marker for a subset of UBCs due to its low expression in granule cells (*Mugnaini et al., 2011*). Tbr2$^+$NeuN$^+$ UBCs in the ML and IGL of the *Nf1*$^{hGFAP}$CKO cerebellum expressed CR$^{high}$ at a significantly lower percentage (36% and 38%) than Tbr2$^{high}$NeuN$^-$ UBCs (67%) typically observed in the IGL (*Figure 5—figure supplement 1A,A'*). Importantly, the absolute number of Tbr2$^+$NeuN$^+$ UBCs was similar in the ML of folium V between the *Nf1*$^{Math1}$CKO and *Nf1*$^{hGFAP}$CKO cerebellum, further supporting the notion that ectopic generation of these UBCs results from loss of cell-autonomous Nf1 function in the glutamatergic neuronal lineages (*Figure 4I'*). Similar to the *Nf1*$^{Math1}$CKO cerebellum, ectopic Tbr2$^+$NeuN$^+$ UBCs were continuously generated in the ML and IGL of the *Nf1*$^{hGFAP}$CKO cerebellum during postnatal stages (arrows, *Figure 5—figure supplement 1B,B',C,C'*), well beyond the developmental window of the generation of control Tbr2$^{high}$NeuN$^-$ UBCs (*Figure 4J,J'*). Together, these observations demonstrate that the Tbr2$^+$ UBCs ectopically generated in both the *Nf1*$^{Math1}$CKO and *Nf1*$^{hGFAP}$CKO cerebellum are different from those previously described in the normal cerebellum in 6 critical aspects: (1) the marker expression, Tbr2$^+$NeuN$^+$ vs Tbr2$^{high}$NeuN$^-$, (2) the low vs high percentage expressing CR$^{high}$, (3) the location, ML and IGL vs IGL, (4) the developmental origin, precursors in the EGL vs RL, (5) the timing of birth, neonatal vs embryonic stages, and (6) the migratory pathway, EGL-ML-IGL vs RL-PWM-IGL. Thus, we suggest the existence of two UBC sublineages. Importantly, the majority of GABARα6-negative neurons were Tbr2$^+$NeuN$^+$ cells, while no significant difference was observed in GABAergic neurons between the control and *Nf1*$^{hGFAP}$CKO cerebellum (*Figure 5—figure supplement 1D,E,E'*). These observations suggest that Tbr2$^+$NeuN$^+$ UBCs are the major contributors to the non-GCs abnormally accumulated in the ML of the *Nf1*$^{hGFAP}$CKO cerebellum (*Figure 1D–D''*).

## Glia-dependent migratory defect of Tbr2$^{high}$NeuN$^-$ UBCs

As observed in the rest of mutant folia, the number of Tbr2$^+$ cells was also significantly increased in the ML of the folium X of the *Nf1*$^{hGFAP}$CKO cerebellum (*Figure 5C,C'*; *Figure 5—figure supplement 1E*). However, the number of Tbr2$^+$ cells was unexpectedly reduced in the IGL of folium X of the *Nf1*$^{hGFAP}$CKO cerebellum concomitant with ectopic presence of Tbr2$^+$ cells in the surrounding WM (*Figure 5C,C'*; *Figure 5—figure supplement 2A*). As in Folium V, the Tbr2$^+$NeuN$^+$ UBC sublineage contributed to almost the entire UBC population in the ML of both the control (97%) and *Nf1*$^{hGFAP}$CKO (99%) folium X (*Figure 5D,D'*). Similarly, Tbr2$^+$NeuN$^+$ UBCs were indeed increased in the IGL of *Nf1*$^{hGFAP}$CKO folium X, and thus the reduction of Tbr2$^+$ UBCs was exclusively attributed to a decreased number of Tbr2$^{high}$NeuN$^-$ UBCs in the IGL (*Figure 5D,D'*). Consistently, UBCs in both the *Nf1*$^{hGFAP}$CKO WM and IGL were the Tbr2$^{high}$NeuN$^-$ sublineage, which was normally located in the IGL of the control cerebellum with a higher percentage expressing CR$^{high}$ (*Figure 5—figure supplement 2B–B''*). More importantly, the total number of Tbr2$^{high}$NeuN$^-$ UBCs was almost identical, when the number of the Tbr2$^{high}$NeuN$^-$ UBCs in the WM and IGL of the *Nf1*$^{hGFAP}$CKO cerebellum was combined to compare with the number of Tbr2$^{high}$NeuN$^-$ UBCs in the IGL of the control cerebellum, suggesting arrested migration in the WM (*Figure 5D–D''*). Consistently, no alteration in the number and distribution of Tbr2$^+$ UBCs was observed at early stages of UBC development before they migrated out of the RL or entered the PWM during perinatal stages (*Figure 5—figure supplement 2C,C'*). Instead, abnormal accumulation of Tbr2$^{high}$NeuN$^-$ UBCs in the *Nf1*$^{hGFAP}$CKO WM was evident at P8, a time point when normal UBCs had already migrated through the PWM and entered the IGL (*Figure 5—figure supplement 2D,D'*). Together, these results suggest that the abnormal accumulation of Tbr2$^{high}$NeuN$^-$ UBCs in the WM of the *Nf1*$^{hGFAP}$CKO cerebellum was caused by arrested migration of UBCs in the PWM on the way from the RL to the IGL during neonatal development (*Englund et al., 2006*).

Importantly, no ectopic Tbr2$^+$ UBCs were observed in the WM of the *Nf1*$^{Math1}$CKO cerebellum, suggesting a non-cell-autonomous mechanism (*Figure 5F*). Consistently, abnormally increased numbers

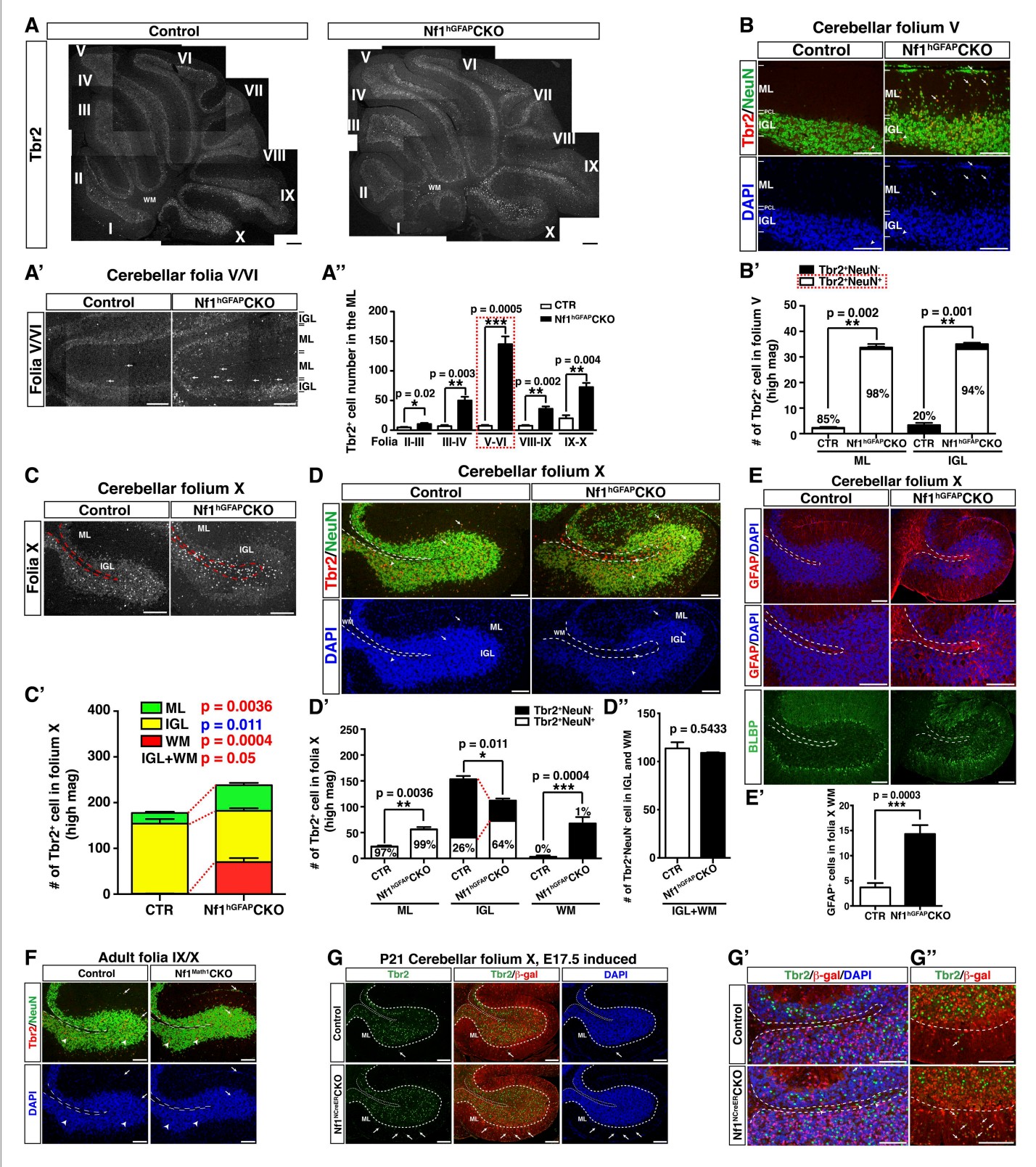

**Figure 5**. Glia-independent and -dependent UBC abnormalities in the *Nf1*[hGFAP]CKO cerebellum. (**A**) The overall expression pattern of Tbr2 in the adult control and *Nf1*[hGFAP]CKO cerebella was compared. High magnification images were merged to provide a high-resolution view. (**A'**) Tbr2 staining in folium V/VI was compared. Arrows point to Tbr2+ cells in the ML. The total number of Tbr2+ cell in the ML of each folium was quantified in (**A''**). (**B**) Sections

*Figure 5. Continued on next page*

*Figure 5. Continued*

were stained for Tbr2/NeuN and imaged at folium V. The number of Tbr2$^+$NeuN$^+$ cells and Tbr2$^+$NeuN$^-$ cells per high magnification field was quantified in (**B'**). The percentage of Tbr2$^+$NeuN$^+$ cells among Tbr2$^+$ cells is illustrated. (**C**) Tbr2 staining in folium X was compared. The total number of Tbr2$^+$ cells in the ML, IGL and WM of folia X was quantified in (**C'**). (**D**) Sections were stained for Tbr2/NeuN and imaged at folium X. The number of Tbr2$^+$NeuN$^+$ cells and Tbr2$^+$NeuN$^-$ cells per high magnification field was quantified in (**D'**) and (**D''**). The percentage of Tbr2$^+$NeuN$^+$ cells is illustrated. (**E**) Sections were stained for GFAP and BLBP and imaged at folium X. The total number of GFAP$^+$ cells in the WM was quantified in (**E'**). (**F**) Cerebellar sections from control and *Nf1*$^{Math1}$CKO were stained for Tbr2/NeuN and compared. (**G–G''**) Control and *Nf1*$^{NcreER}$CKO mice were TM-induced at E17.5 and analyzed at P21. Tbr2/β-gal staining was imaged at folium X (**G**) and high magnification views of the white matter (**G'**) and ventral folium X (**G''**) were compared. Dashed lines mark the border between the WM and IGL in folium X. Of note, ectopic Tbr2 cells in the WM of *Nf1*$^{NcreER}$CKO were exclusively β-gal-negative (*Nf1*-wildtype) (**G'**). In contrast, the increase of Tbr2$^+$ cells in the mutant ML was mostly contributed by Tbr2$^+$β-gal$^+$ cells (*Nf1*-deficient) (arrows, **G''**). Arrows label co-localizing cells and arrowheads label non-co-localizing cells. All the quantification data are presented as mean ± SEM. DAPI labels the nuclei. Scale bars: 50 μm.

The following figure supplements are available for figure 5:

**Figure supplement 1**. The ectopic accumulation of Tbr2$^+$ UBC in the ML of *Nf1*$^{hGFAP}$CKO cerebella occurs during postnatal development.

**Figure supplement 2**. A subpopulation of UBCs in the *Nf1*$^{hGFAP}$CKO cerebella fail to migrate into the IGL and are ectopically present in the WM during postnatal development.

of GFAP$^+$BLBP$^-$ astrocytes were observed in the enlarged WM of the adult *Nf1*$^{hGFAP}$CKO cerebellum, raising the possibility that excess astrocytes in the WM could impede the migration of UBCs from the WM to the IGL (*Figure 5E,E'*). In the TM-induced *Nf1*$^{NCreER}$CKO cerebellum where the number of glial cells was increased specifically in the posterior folia, a subpopulation of Tbr2$^{high}$ UBCs, all of which were wild-type for *Nf1* (not labeled by β-gal expression), ectopically accumulated in the WM of folium X (arrowheads, *Figure 5G,G'*). Of note, excess Tbr2$^+$ UBCs in the ML of the TM-induced *Nf1*$^{NCreER}$CKO cerebellum appears to result from overproduction of *Nf1*-deficient Tbr2$^+$β-gal$^+$ UBCs—a cell-autonomous mechanism as described in the *Nf1*$^{Math1}$CKO cerebellum (*Figure 5G,G''*). These results suggest that overproduction of astrocytes in the WM of the *Nf1*$^{hGFAP}$CKO and *Nf1*$^{NCreER}$CKO cerebella produces a physical barrier that exerts non-cell-autonomous effect, arresting migration of Tbr2$^{high}$NeuN$^-$ UBCs in the WM.

## MEK inhibition rescues neuronal defects in the developing *Nf1*$^{Math1}$CKO cerebellum

To develop a therapeutic strategy to rescue cerebellar defects caused by *Nf1* inactivation, we analyzed three major downstream effectors of Ras-mediated signaling pathways: Erk/MAPK, PI3K/Akt and mTORC1. Although Western blot analysis showed a significant reduction of neurofibromin expression, Erk/MAPK was the only downstream signaling pathway that was consistently hyperactivated in *Nf1*$^{Math1}$CKO cerebella (*Figure 6A*). Consistent with the specificity of Math1-cre transgene, cell cultures with enriched P6 GCPs showed almost a complete loss of neurofibromin expression with a robust increase of Erk activation (*Figure 6A'*). A high level of phosphorylated Erk (p-Erk) expression was observed in a subset of IGL cells, while a low level in the ML and a complete absence of p-Erk in the EGL was found in the control cerebellum at P8 (*Figure 6B*). The pattern of p-Erk expression is consistent with a possible role of a high level of Erk signaling in neuronal differentiation. A robust increase of p-Erk expression was observed in the P8 *Nf1*$^{Math1}$CKO cerebellum including the inner edge of the EGL (pre-migratory zone), the ML, and IGL (*Figure 6B*). Importantly, the NeuN$^+$ neurons abnormally accumulated in the ML of the P8 *Nf1*$^{Math1}$CKO cerebellum expressed a high level of p-Erk (arrows, *Figure 6B*). These results demonstrate that Nf1 is a major regulator that suppresses Erk signaling in migrating GCs in the pre-migratory EGL and migratory zone (ML) of the developing cerebellum. To test whether hyperactive Erk signaling is responsible for *Nf1*-deficient GCPs to undergo premature differentiation and ectopically adopt a UBC fate, we treated newborn control and *Nf1*$^{Math1}$CKO pups with a MEK inhibitor (PD0325901) or vehicle from P0.5 to P21 for 3 weeks (*Sebolt-Leopold and Herrera, 2004*; *Wang et al., 2012*). This 3-week treatment protocol exhibited no adverse effect on normal cerebellar development, but reduced p-Erk expression in both MEKi-treated control and *Nf1*$^{Math1}$CKO cerebella (data not shown). Strikingly, MEK inhibition completely eliminated ectopic Tbr2 expression in the ML of MEKi-treated *Nf1*$^{Math1}$CKO cerebella (*Figure 6C*). MEK inhibition consistently

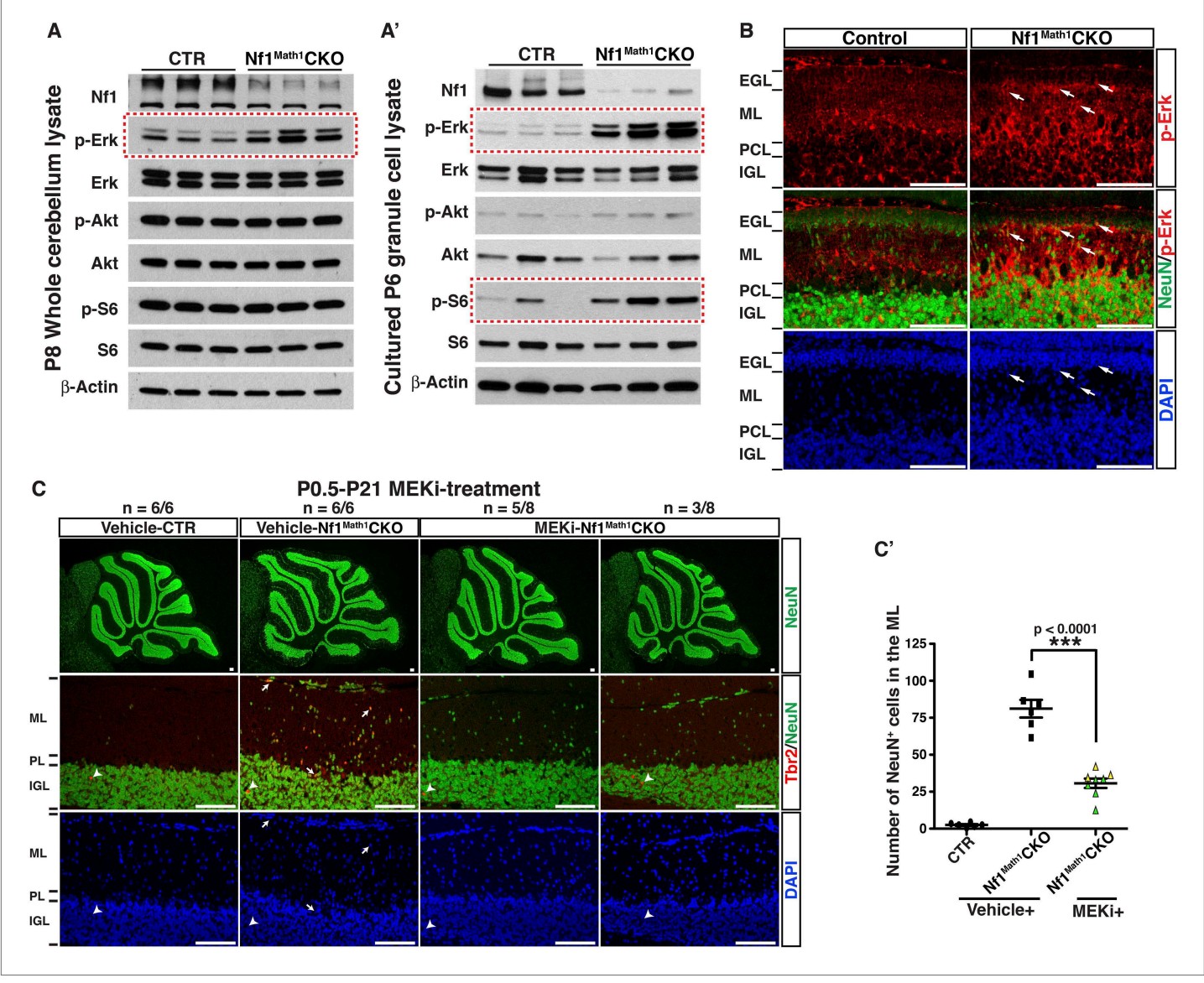

**Figure 6**. Glia-independent neuronal defects in the *Nf1*[Math1]CKO cerebellum are rescued by P0.5–P21 MEKi-treatment. Western blot analysis was performed on P8 whole cerebellar lysates (**A**) or P6 cultured GCP lysates (**A'**) from control and *Nf1*[Math1]CKO mice. (**B**) Cerebellar sections from P8 control and *Nf1*[Math1]CKO mice were stained for NeuN and p-Erk. Arrows highlight the NeuN+p-Erk+ cells in the mutant EGL and ML. (**C**) Control and *Nf1*[Math1]CKO mice were treated with vehicle or MEKi (5 mg/kg) from P0.5–P21 and analyzed at P21. Cerebellar sections were stained for Tbr2/NeuN, and imaged at low magnification (upper panels) and folium V (lower panels). (**C'**) The number of NeuN+ cells in the ML was quantified and compared. Of note, the yellow triangles represent 3 MEKi-treated mutant cerebella that still displayed cell clusters near the pial surface. Individual data points are presented, as well as mean ± SEM for each group. DAPI labels the nuclei. Scale bars: 50 μm.

The following figure supplement is available for figure 6:

**Figure supplement 1**. Rapamycin treatment during neonatal stages does not rescue the neuronal defects in the *Nf1*[Math1]CKO cerebellum, but causes adverse effects.

reduced over 60% of abnormally accumulated neurons in the ML of the *Nf1*[Math1]CKO cerebellum (*Figure 6C,C'*). Since an increase of p-S6 staining was also identified in purified GCPs from the P6 *Nf1*[Math1]CKO cerebellum compared to controls (*Figure 6A'*), we treated control and *Nf1*[Math1]CKO mice with rapamycin, an mTORC1 inhibitor (*Guertin and Sabatini, 2009*). Rapamycin treatment had no benefit on neuronal defects in the *Nf1*[Math1]CKO cerebellum, but caused adverse effects on neuronal

migration in both the control and mutant cerebellum (*Figure 6—figure supplement 1*). Together, these results demonstrate that Nf1-mediated MEK/Erk inhibition is the major mechanism underlying the suppression of a subset of GCPs that otherwise would undergo premature differentiation and ectopically adopt a UBC fate in the developing cerebellum.

## Short- and long-term benefits of MEKi treatment on cerebellar and motor defects of *Nf1*[hGFAP]CKO mice

Based on the success of MEK inhibition on the developing *Nf1*[Math1]CKO cerebellum, we sought to investigate whether MEK inhibition could also rescue the more severely affected *Nf1*[hGFAP]CKO cerebellum. The MEKi-treated *Nf1*[hGFAP]CKO mice displayed a varying degree of therapeutic responses in glial and neuronal defects in the developing cerebellum as well as their general behaviors (*Video 1*). Of 14 MEKi-treated *Nf1*[hGFAP]CKO cerebella analyzed, 9 mice (good-responders or G-responders) exhibited a significant rescue in the number of neurons abnormally accumulated in the ML of the developing cerebellum at a level similar to what was achieved in the MEKi-treated *Nf1*[Math1]CKO mice (green color, *Figure 7A–A''*). However, the remaining 5 of the MEKi-treated *Nf1*[hGFAP]CKO mice (poor-responders or P-responders) only exhibited marginal improvement in the neuronal defect (yellow color, *Figure 7A–A''*). Importantly, G- and P-responders could be readily distinguished by the presence or absence of rescues (to the control levels) in the number of BG cells, alignment of the BG scaffold, the number of mispositioned Purkinje cells and arrested UBCs in the WM (*Figure 7A,B,B',C,C'*). It should be noted that both G-responders and P-responders exhibited a complete rescue of ectopic Tbr2 expression in the ML of the MEKi-treated *Nf1*[hGFAP]CKO mice (*Figure 7D,D'*). Therefore, the fundamental difference in the cerebella of G- and P-responders lies in the presence or absence of rescue in glia-dependent cerebellar defects, which could result from inconsistent inhibition of much higher levels of hyperactive Erk signaling in *Nf1*-deficient glial lineage cells than that in *Nf1*-deficient immature neurons in the ML (*Figure 7E*; *Figure 7—figure supplement 1A,B*). Consistently, when we increased the dose of MEKi from 5 mg/kg to 20 mg/kg to lactating females, more consistent rescues were observed in the cerebella of MEKi-treated *Nf1*[hGFAP]CKO pups (*Figure 7F*; *Figure 7—figure supplement 2*).

We next sought to determine whether the transient P0.5–P21 treatment protocol could exert a long-term and sustained therapeutic benefit. When analyzed at P45–P60—1 month after MEKi treatment was terminated, MEKi-treated *Nf1*[hGFAP]CKO G-responders exhibited the rescue effects on both glial and neuronal defects, while P-responders continued to have defects similar to those observed after acute treatment at P21 (*Figure 8A,A'*). To investigate whether the rescue of the cerebellar cellular and structural defects could improve motor functions, we performed a Rotarod test on control and *Nf1*[hGFAP]CKO mice for 4 consecutive days immediately after 3-week MEKi treatment. The MEKi-treated *Nf1*[hGFAP]CKO mice stayed significantly longer on the Rotarod compared to vehicle-treated mutant mice (*Figure 8B*). Moreover, while vehicle-treated *Nf1*[hGFAP]CKO mice did not improve motor performance over the 4-day training course, the MEKi-treated mutant mice exhibited a significant improvement similar to controls, suggesting MEKi treatment also enhanced motor learning (*Figure 8B*). Importantly, the improved motor performance was maintained at least 1 month after MEKi treatment was terminated, though the rescue in motor learning was lost (*Figure 8B'*; *Video 2*). G-responders stayed significantly longer on Rotarod than P-responders, providing a nice correlation between recues in cerebellar developmental defects and motor functions of MEKi-treated *Nf1*[hGFAP]CKO mice (*Figure 8C*). Together, these results demonstrate that transient MEKi treatment during neonatal stages prevents cerebellar developmental defects and provides long-term benefits on motor functions of *Nf1*[hGFAP]CKO mice.

## Discussion

By utilizing four GEM models, we have identified essential roles of *Nf1* tumor suppressor gene during early, but not late stages of cerebellar development. Mechanistically, Nf1 is required for the formation of the BG monolayer and scaffold by controlling astrocyte precursor proliferation at a specific precursor stage, but dispensable for BG cells during late stages of cerebellar development and adulthood. Similarly, Nf1 indirectly regulates glutamatergic neuronal migration by controlling the timely differentiation and suppressing ectopic UBC differentiation of GCPs during postnatal cerebellar development. The timely requirement of Nf1 during cerebellar development allows us to develop a therapeutic prevention strategy for treating these developmental disorders with long-term behavioral benefits.

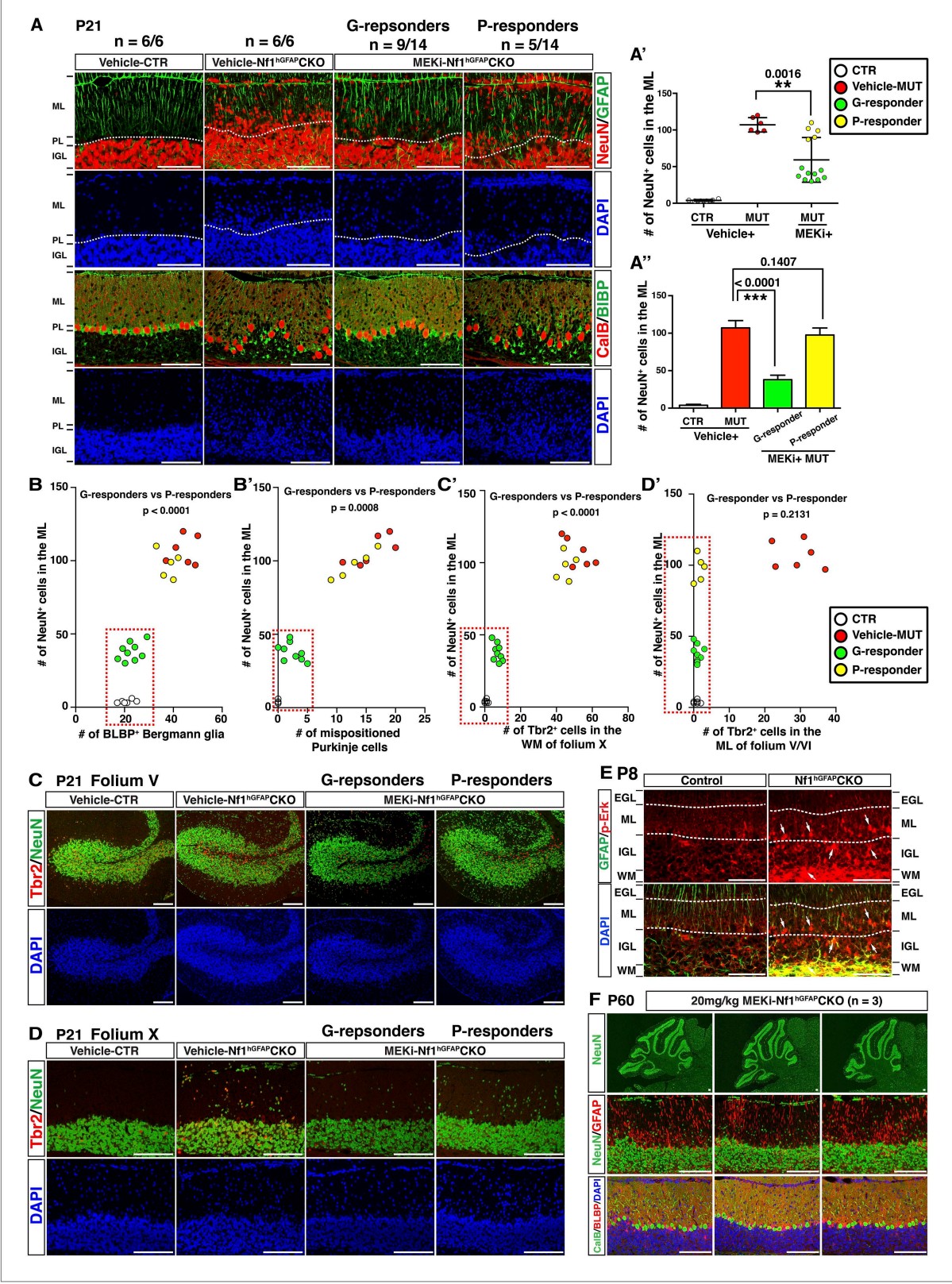

**Figure 7.** MEKi treatment rescues glia-dependent cerebellar defects in the *Nf1*hGFAPCKO cerebellum. (**A**) Cerebellar sections of vehicle- and MEKi-treated (5 mg/kg) control and *Nf1*hGFAPCKO mice were stained for GFAP/NeuN and CalB/BLBP and imaged at folium V. 9 out 14 MEKi-treated mutant mice (G-responder) exhibited a significant improvement, although the remaining 5 (P-responder) still had large number of cells in the EGL/ML and displayed

*Figure 7. Continued on next page*

*Figure 7. Continued*

severe laminar disruption. The number of NeuN⁺ cells in the ML was quantified and compared in (**A′**). For MEKi-treated mutants, green and yellow color represents G-responders and P-responders, respectively. G-responders and P-responders were separated into two groups and compared in (**A″**). (**C**, **D**) Cerebellar sections were stained for Tbr2/NeuN and imaged at folia V and X. (**B**, **B′**, **C′**, **D′**) The correlation between the number of NeuN⁺ cells in the ML, and the number of BLBP⁺ BG cells (**B**), mispositioned Purkinje cells (**B′**), ectopic Tbr2⁺ cells in the WM (**C′**) and ML (**D′**) was plotted and compared. (**E**) Sections of P8 control and *Nf1*[hGFAP]CKO cerebella were stained for GFAP and p-Erk. Arrows highlight the GFAP⁺p-Erk⁺ cells in the ML, IGL and WM of the *Nf1*[hGFAP]CKO cerebellum. (**F**) Three *Nf1*[hGFAP]CKO mice treated with MEK inhibitor at 20 mg/kg from P0.5–P21 were analyzed at P60. GFAP/NeuN and CalB/BLBP staining show consistent rescue of both neuronal and glial defects. Individual data points are presented, as well as mean ± SEM for each group. DAPI labels the nuclei. Scale bars: 50 μm.

The following figure supplements are available for figure 7:

**Figure supplement 1**. Ras/Erk signaling is differentially activated in neuronal and glial precursors of the developing *Nf1*[hGFAP]CKO cerebellum.

**Figure supplement 2**. High-dose MEKi-treatment produces more consistent phenotypic rescue.

## Nf1 regulates astrocyte proliferation at a specific precursor stage

Astrocytes, the most abundant cells in the brain, play diverse roles in both the developing and mature brain (*Barres, 2008*; *Molofsky et al., 2012*; *Freeman and Rowitch, 2013*). Compared to the other two major CNS cell types, neurons and oligodendrocytes, the identity of astrocyte precursors remains poorly understood. This is largely due to a lack of known markers that can reliably define astrocytes at distinct developmental stages (*Molofsky et al., 2012*; *Freeman and Rowitch, 2013*). BG cells have a unique and well-defined migratory pathway coupled with differentiation that allows us to define the role of Nf1 in this particular astrocyte lineage at different developmental stages (*Buffo and Rossi, 2013*). Before becoming terminally differentiated in the PC/BG monolayer, BG precursors proliferate at three distinct stages of differentiation that can be distinguished by different anatomical locations during development (*Figure 8D*). These three stages include: Stage I, RG/early-precursor stage in the IV-VZ; Stage II, BG intermediate-precursor stage in the PWM; Stage III, BG late-precursor stage in the rudimentary PC/BG layer. In the *Nf1*[hGFAP]CKO cerebellum, increased proliferation was not observed in E17.5 IV-VZ RG cells (Stage I), or in BG late-precursors following migration into the rudimentary PC/BG layer in P0.5 anterior folia, or throughout P8 folia (Stage III). In contrast, a significant increase of proliferation index was specifically identified in *Nf1*-deficient BG intermediate-precursors in the E17.5 PWM and P0.5 posterior folia (Stage II), leading to an increased number of both total BLBP⁺ and proliferating BLBP⁺Ki67⁺ cells. These observations suggest that Nf1 plays a robust role in constraining BG proliferation specifically at an intermediate precursor stage (Stage II), but not during earlier or later stages (Stage I or Stage III). This conclusion is further confirmed by acute inactivation of *Nf1* in the E17.5 developing cerebellum using an inducible *Nf1*[NCreER]CKO model. Thus, the temporal and regional phenotypic differences upon *Nf1* loss observed in the BG lineage can be strictly attributed to differences in the differentiation status of this particular astrocyte lineage. Moreover, *Nf1*-deficient astrocytes in the developing WM also exhibited increased proliferation. Since gliogenesis occurs at different time points in different regions of the developing brain, this 'differentiation-based' model might provide an explanation for conflicting results on the role of Nf1 in neural stem and progenitor cells among previously published studies (*Dasgupta and Gutmann, 2005*; *Zhu et al., 2005*; *Hegedus et al., 2007*; *Becher and Holland, 2010*; *Lee da et al., 2010*; *Lee da et al., 2012*; *Wang et al., 2012*). It should be noted that region-specific mechanisms underlying glial pathologies induced by *Nf1* loss do exist within the dorsal brain. While increased proliferation in *Nf1*-deficient astrocyte precursors appears to be responsible for the increase of glial cells in the dorsal hindbrain (e.g., cerebellum), altered fate-specification of *Nf1*-deficient neural stem and progenitor cells in the subventricular zone (SVZ) of the lateral ventricle causes the increase of glial cells in the corpus callosum by promoting gliogenesis at the expense of neurogenesis in the olfactory bulb (*Wang et al., 2012*).

It is not entirely clear why Nf1 plays a critical role in suppressing astrocyte proliferation at a specific precursor stage. Loss of function studies in mice have shown that receptor tyrosine kinase (RTK) signaling, particularly fibroblast growth factor receptor (FGFR) signaling, is essential for the transition from RG to BG cells (*Lin et al., 2009*; *Muller Smith et al., 2012*; *Meier et al., 2014*).

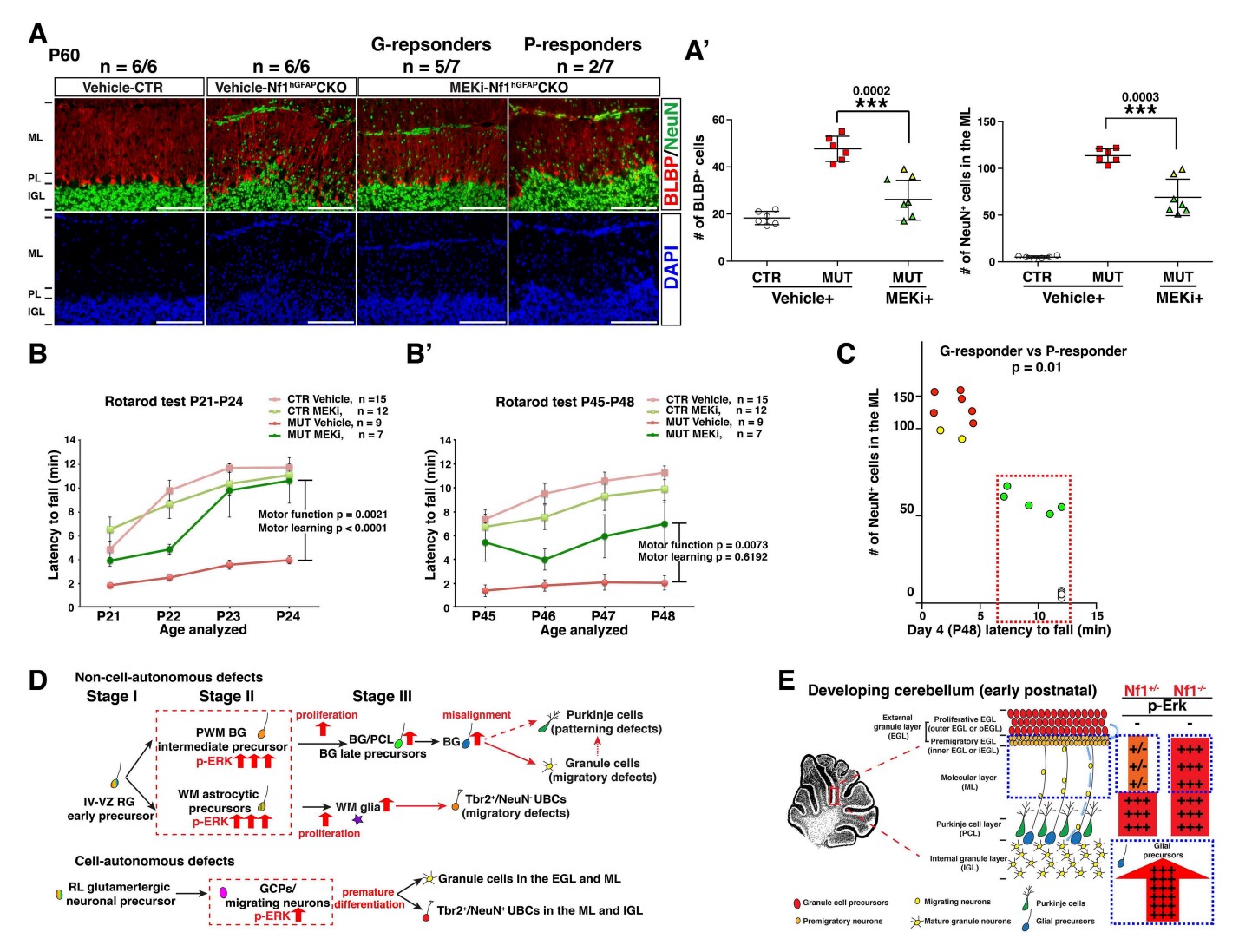

**Figure 8**. Transient MEKi treatment provides long-term rescue of the cerebellar defects and improves the motor performance of *Nf1*[hGFAP]CKO mice. (**A**) Control and *Nf1*[hGFAP]CKO mice were treated with vehicle or MEK inhibitor (MEKi) from P0.5–P21 and analyzed for BLBP/NeuN staining between P50 and P60. The number of NeuN+ cells in the ML and BLBP+ BG cells were quantified, respectively (**A'**). (**B**, **B'**) Rotarod tests were performed on transiently vehicle- and MEKi-treated control and *Nf1*[hGFAP]CKO mice for consecutive days from P21–P24 (**B**) or P45–P48 (**B'**) and the duration time for the mice to stay on the Rotarod was documented and compared. The correlation between the number of NeuN+ cells in the ML and duration time on the Rotarod at P48 was plotted in (**C**). (**D**) A model is proposed to summarize cell-autonomous and non-cell-autonomous defects in *Nf1*[hGFAP]CKO cerebellum and the underlying mechanisms. See main text. (**E**) A schematic diagram illustrates the differential Erk activation in the P8 control and *Nf1*[hGFAP]CKO cerebella. In controls, p-Erk expression is mostly restricted in the IGL neurons and a subset of WM glia, while in mutants, high level of p-ERK expression is identified in migrating neurons in the ML and the majority of WM glia. Note that in *Nf1*[hGFAP]CKO cerebella, p-ERK level is much higher in glia compared to neurons. Individual data points are presented, as well as mean ± SEM for each group. DAPI labels the nuclei. Scale bars: 50 μm.

Most strikingly, loss of Shp2, also known as Ptpn11 (Protein tyrosine phosphatase non-receptor type 11), completely blocks the transition from RG to BG cells, which can be rescued by constitutively active MEK (*Li et al., 2014*). These studies suggest that activation of Shp2-dependent FGF/MEK/Erk signaling is essential for the transition from RG to BG cells during cerebellar development (from Stage I to II). Accordingly, our study suggests that Erk signaling must be constrained by pathway inhibitors such as Nf1 to ensure the generation of the correct number of BG cells. Consistently, the most robust Erk/MAPK activation upon Nf1 loss was observed in astrocyte precursors at Stage II and early Stage III (*Figure 8E*).

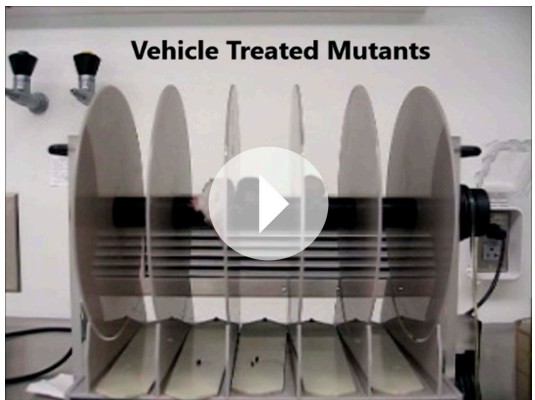

**Video 2**. Rotarod testing. Representative P48 control and *Nf1*ʰᴳᶠᴬᴾCKO mice were videotaped on the final (fourth) day of rotarod performance testing. During the test, the vehicle-treated mutants had difficulty walking in a constant forward direction, rotated with the rotarod, and had frequent slippage with difficulty maintaining a grip on the rotarod ultimately resulting in a fall before 5 min. The MEKi-treated mice all had improved performance on the rotarod test, staying on the rotarod for the entire duration, but had a variable response to the treatment, as some of the MEKi-treated mice continued to show evidence of motor dysfunction (e.g., more likely to cling onto the rotarod instead of walking). The video includes clips from different time points and speeds during the rotarod test as a representation of the mouse performance during the task.

## Glia-dependent and –independent Nf1 functions in glutamatergic interneurons

Our study provides several lines of evidence supporting the notion that the correct number of astrocytes is essential for the formation of the PC/BG monolayer and the BG scaffold, and subsequently glutamatergic interneuronal migration. First, in the *Nf1*ʰᴳᶠᴬᴾCKO cerebellum, the severity of glial and neuronal defects is not uniform, but highly correlated with the excess number of BG cells in the affected areas. Indeed, the capacity of *Nf1*-deficient BG cells to extend their radial fibers appeared relatively uncompromised even for those projected from the areas outside of the PC/BG layer. Second, considering the time required for TM induced, Cre-mediated *Nf1* deletion at E17.5 and the half-life of neurofibromin protein, we expect that a complete loss of Nf1 would be fully achieved in targeted cells around P0.5 when BG cells in the anterior folia (I–V) are at Stage III, while BG cells in the posterior folia (IX and X) are still proliferating in the PWM at Stage II. Consistently, a significant increase of *Nf1*-deficient BG cells was observed in the posterior folia (IX, X), but not in the anterior folia of the same *Nf1*ᴺᶜʳᵉᴱᴿCKO cerebellum. Moreover, failed formation of the PC/BG monolayer, misalignment of the BG scaffold, and abnormal accumulation of both *Nf1*-intact and *Nf1*-deficient neurons in the ML were only observed in the posterior folia of the E17.5 TM-induced *Nf1*ᴺᶜʳᵉᴱᴿCKO cerebellum. It should be noted that glial and neuronal defects in the posterior folia of the E17.5 TM-induced *Nf1*ᴺᶜʳᵉᴱᴿCKO cerebellum are significantly less severe than those observed in the *Nf1*ʰᴳᶠᴬᴾCKO cerebellum. This is likely due to a relatively greater window for the expansion of *Nf1*-deficient glial precursors in the *Nf1*ʰᴳᶠᴬᴾCKO cerebellum (from E12.5 on) than E17.5 TM-induced *Nf1*ᴺᶜʳᵉᴱᴿCKO cerebellum. Third, transient treatment of MEKi exhibited a robust long-term rescue of glial and neuronal defects in the cerebella of G-responders in which neurons and glia were still deficient for *Nf1* after the treatment. Taken together, these results suggest that as long as the excess number of glia is therapeutically fixed during a specific window of cerebellar development, a long-term rescue could be achieved, as Nf1 is relatively dispensable for glial cell proliferation during late stages of development and adulthood.

Our study also shows that the correct number of astrocytes in the developing WM of the cerebellum is essential for the migration of UBCs from the PWM to the IGL. This neuronal migratory defect appears to result from loss of a non-cell-autonomous Nf1 function in glial cells. This conclusion is based on the observations that (1) neuron-specific *Nf1*ᴹᵃᵗʰ¹CKO cerebellum exhibited no such defect in the WM, and (2) *Nf1*-intact UBCs were arrested in the WM of the E17.5 TM-induced *Nf1*ᴺᶜʳᵉᴱᴿCKO cerebellum where excess astrocytes were present. Furthermore, the migratory defect of UBCs was completely rescued in the transiently MEKi-treated cerebella of G-responders, which contained a relatively normal number of glial cells. Together, our study demonstrates that Nf1 not only plays a critical role in suppressing neoplastic transformation of glial cells, but also is essential for controlling the generation of the correct number of normal astrocytes—which is required for the migration of both types of glutamatergic interneurons (GCs and UBCs) during cerebellar development.

Analysis of the neuron-specific *Nf1*ᴹᵃᵗʰ¹CKO cerebellum supports the existence of two UBC sublineages. One UBC sublineage characterized by Tbr2ʰⁱᵍʰNeuN⁻ is relatively well studied, which exit cell cycles during perinatal stages and migrate from the RL via the PWM to the IGL (*Englund et al., 2006*;

*Mugnaini et al., 2011*). Although Nf1-regulated glial number is essential for the migration of this UBC sublineage through the PWM, Nf1 is not required for the generation and migration of the Tbr2$^{high}$NeuN$^-$ UBC sublineage. In contrast, the other rare Tbr2$^+$NeuN$^+$ UBC sublineage appears to be suppressed by Nf1-regulated Erk signaling in GCPs in a cell-autonomous mechanism. Thus, our study demonstrates that Nf1 is indirectly required for GC migration by suppressing GCPs to undergo premature terminal differentiation and ectopic differentiation to Tbr2$^+$NeuN$^+$ UBCs in the EGL and ML. It is still possible that *Nf1* loss causes abnormal migration of the Tbr2$^{high}$NeuN$^-$ sublineage from the RL to the EGL, and then switches them to the Tbr2$^+$NeuN$^+$ UBC sublineage in the EGL. However, no Tbr2$^+$ cells were found in the EGL of the mutant cerebellum at E17.5 or P0.5, when Tbr2$^{high}$NeuN$^-$ UBCs migrate out of the RL into the PWM (*Figure 5—figure supplement 2C,C'*). This observation argues against the possibility of abnormal migration of Tbr2$^{high}$NeuN$^-$ UBCs into the EGL. Indeed, *Nf1* loss has little or no impact on the generation, migration and total number of Tbr2$^{high}$NeuN$^-$ UBCs in the mutant cerebellum. Furthermore, in contrast to the precursors of Tbr2$^{high}$NeuN$^-$ UBCs that exit cell cycles during perinatal stages, *Nf1*-deficient precursors of Tbr2$^+$NeuN$^+$ UBCs continue proliferating in the EGL at least until P8 at the peak of GCP proliferation. Together, these observations suggest that Nf1 plays a specific role in the Tbr2$^+$NeuN$^+$ UBC sublineage by controlling the fate-specification of GCPs in differentiating into GCs vs Tbr2$^+$NeuN$^+$ UBCs. The involvement of UBCs in cerebellar vestibular functions raises the possibility that the UBC defects in these *Nf1* GEM models could contribute to the impairments of motor and balance observed in individuals with NF1 (*Mugnaini et al., 2011*; *Champion et al., 2014*).

## Clinical implications

Our previous study showed that MEKi could be delivered to neonatal pups through milk from MEKi-treated lactating females (*Wang et al., 2012*). This 'MEKi in milk' treatment protocol allows a low dose of MEKi to get into pups without introducing any physical distress. When lactating females received MEKi at 5 mg/kg (body weight), a mild 30–40% of Erk inhibition was observed in the brains of *Nf1*$^{hGFAP}$CKO pups (MB, EJ and YZ, unpublished observations). Strikingly, this mild Erk inhibition robustly and consistently rescued developmental defects in the cerebella of 100% of MEKi-treated *Nf1*$^{Math1}$CKO mice, as well as over 60% of the MEKi-treated *Nf1*$^{hGFAP}$CKO mice (G-responders). It should be noted that the remaining P-responders are not 'non-responders', as these MEKi-treated *Nf1*$^{hGFAP}$CKO mice still exhibited a complete rescue of ectopic Tbr2 expression in their cerebella, and consistent rescues in the defects in the corpus callosum and SVZ, as described previously (*Wang et al., 2012*). The inconsistent rescue in the glia-dependent defects observed in the cerebella of MEKi-treated *Nf1*$^{hGFAP}$CKO mice is likely due to the extremely high levels of Erk activation in *Nf1*-deficient glial cells in the developing cerebellum compared to other cells. Indeed, when the dose of MEKi was increased to 20 mg/kg, a more consistent rescue in cerebellar defects was observed in the cerebella of the MEKi-treated *Nf1*$^{hGFAP}$CKO mice (*Figure 7F*). More importantly, this 'MEKi in milk' treatment protocol provides robust long-term rescues on cerebellar developmental defects and motor performance. It should be emphasized that this transient MEKi treatment protocol also rescued structural defects in the other regions of the *Nf1*$^{hGFAP}$CKO brain including the enlarged corpus callosum, a brain structural defect associated with severe cognitive impairments in a subset of NF1 patients (*Wang et al., 2012*). Thus, motor performance improvement by this protocol is a collective reflection of rescues of multiple NF1-associated brain structural defects including those in the cerebellum and corpus callosum. Given that cognitive and motor impairments are often co-morbidities in children with NF1 (*Champion et al., 2014*), our findings suggest that this transient 'MEKi in milk' protocol can serve as a common prevention strategy for multiple NF1-associated neurodevelopmental disorders.

## Materials and methods

### Control and mutant mice

Three independent Cre transgenic strains under the control of hGFAP, Math1 and L7 promoters were utilized to target an *Nf1* mutation into different cell lineages during cerebellar development. The control mice for *Nf1*$^{hGFAP}$CKO mice are a pool of phenotypically indistinguishable mice: hGFAP-cre-;*Nf1*$^{flox/flox}$, hGFAP-cre−;*Nf1*$^{flox/+}$ and hGFAP-cre+;*Nf1*$^{flox/+}$. The control mice with similar genetic configurations were used for *Nf1*$^{Math1}$CKO and *Nf1*$^{L7}$CKO mice. All the CKO mice have genotypes of Cre+;*Nf1*$^{flox/flox}$. *Nf1*$^{hGFAP}$CKO, *Nf1*$^{Math1}$CKO, *Nf1*$^{L7}$CKO and control mice were maintained in the

mixed backgrounds of C57Bl6, 129Svj and FVB, which improved overall health and lifespan of *Nf1*[hGFAP]CKO mice. Age- and littermate-matched control and mutant mice were used for analysis. Tamoxifen inducible strain *Nf1*[NcreER]CKO is described in later sections along with the tamoxifen administration protocol. All mice in this study were cared for according to the guidelines that were approved by the Animal Care and Use Committees of the University of Michigan at Ann Arbor, MI and the Children's National Medical Center in Washington, DC.

## MEK inhibitor treatment

MEK inhibitor (MEKi, PD0325901, Sigma-Aldrich, St. Louis, MO) was dissolved in DMSO at a concentration of 25 mg/ml and resuspended in vehicle (0.5% hydroxypropyl methyl-cellulose with 0.2% Tween80, Sigma-Aldrich) at a concentration of 1 mg/ml. The solution was administered by oral gavage at the dosage of 5 mg/kg or 20 mg/kg (body weight) daily to lactating females for the treatment of P0.5–P21 mice. MEKi-treated mice were collected and compared to littermate control mice and vehicle-treated *Nf1*[hGFAP]CKO and *Nf1*[Math1]CKO mice.

## Tissue preparation and histological analysis

Both paraffin and frozen sections were utilized for histological analysis. Control and mutant littermates at various time points were perfused with Phosphate buffered saline (PBS) followed by 4% paraformaldehyde (PFA) (Sigma-Aldrich). Brains were divided into two hemispheres along the midline and each hemisphere was processed for either frozen blocks (for in situ hybridization and X-gal staining, see below) or paraffin-embedded blocks. For paraffin sections, brains were dissected and post-fixed in 4% PFA overnight at 4°C. Fixed brains were sagittally sectioned at 5 μm thickness. Slides from histologically comparable positions were stained by hematoxylin and eosin (H&E) and imaged with a light microscope (Olympus BX51, Olympus America Inc., Center Valley, PA).

## Immunohistochemistry and immunofluorescence

Immunohistochemistry was performed on paraffin sections as previously described (*Wang et al., 2009*). The visualization of primary antibodies was performed with the avidin-biotin horseradish peroxidase system (Vectastain ABC kit, Vector Labs, Burlingame, CA). The dilutions of primary antibodies used on paraffin sections in this study were: BLBP (1:2000, rabbit, a kind gift from Dr. Todd Anthony), Calbindin (1:1000, mouse, Sigma-Aldrich), GFAP (1:2000, rabbit, DAKO, Carpinteria, CA) and BrdU (1:1000, rat, Abcam, Cambridge, MA). Immunofluorescence was performed on both paraffin and frozen sections. Primary antibodies were visualized by Cy2 (or Alexa 488), Cy3 (or Alexa 555) and Cy5 (or Alexa 647)-conjugated secondary antibodies (1:200, Cy2/Cy3/Cy5, Jackson Immunoresearch, West Grove, PA; 1:400, Alexa 488/555/647, Invitrogen, Grand Island, NY). The dilutions of primary antibodies used in this study were as follows: GFAP (1:2000, mouse, BD Pharmingen, San Jose, CA), GFAP (1:2000, rabbit, DAKO), NeuN (1:400, mouse, Millipore, Billerica, MA), BLBP (1:500, rabbit, a gift from Nathaniel Heintz), Calbindin (1:1000, mouse, Sigma-Aldrich), GABARα6 (1:500, rabbit, Millipore), BrdU (1:1000, rat, Abcam), Ki-67 (1:500, mouse, BD Pharmingen), Tbr2 (1:1000, rabbit, Abcam), Calretinin (1:1000, mouse, Millipore), MAP2 (1:1000, mouse, Sigma-Aldrich), P27 (1:250, rabbit, Santa Cruz Biotechnology, Dallas, TX), β-gal (1:1000, rabbit, 5 prime-3 prime, Gaithersburg, MD), β-gal (1:3000, chicken, Abcam), p-Erk (1:200, rabbit, Cell Signaling, Beverly, MA), p-S6 (1:1000, rabbit, Cell Signaling). Sections were examined under a light/fluorescence microscope (Olympus BX51).

## In situ hybridization

Brains were removed immediately after decapitation, frozen in a bed of crushed dry ice, sectioned in a sagittal plane at 14 μm with a cryostat, mounted on RNase-free slides, and treated with 4% paraformaldehyde, and incubated in 0.1 M triethanolamine with acetic anhydride. Radioactive riboprobes were synthesized from template DNA and labeled by $S^{35}$-UTP. For hybridization, an 800 bp Math1 antisense probe was made from a plasmid pLA1-HindIII containing 600 bp of the Math1-coding region and approximately 200 bp of the 5'-flanking sequence (*Helms and Johnson, 1998*). The Zic1 antisense probe was a 690-bp XhoI–SalI segment from the mouse Zic1 cDNA (*Aruga et al., 1994*). After hybridization, slides were dipped in photographic emulsion, counterstained, dehydrated and cover-slipped. Tissue were visualized and photographed with darkfield using an Olympus BX51 microscope. Detailed protocol was described previously (*Martin-Zanca et al., 1990*).

## Tamoxifen administration in Nestin-cre[ER] inducible mice

Tamoxifen (Sigma-Aldrich) was dissolved in corn oil (Sigma-Aldrich) at a concentration of 20 mg/ml and stored at −20°C. Pregnant female mice were intra-peritoneal (IP) injected once with tamoxifen at a dosage of 100 µg/g body weight at E17.5. Their offspring carrying Nestin-cre[ER] transgene were collected at P21 for analysis. Nestin-cre[ER]+; *Nf1*[flox/+]; R26[LacZR] mice were used as controls and Nestin-cre[ER]+; *Nf1*[flox/flox]; R26[LacZR] mice were used as mutants (*Nf1*[NcreER]CKO).

## X-gal staining

The R26[LacZR] allele was introduced to the control, *Nf1*[hGFAP]CKO, *Nf1*[Math1]CKO, *Nf1*[L7]CKO and *Nf1*[NcreER]CKO mice as a reporter to monitor Cre-mediated recombination. Dissected brains were prepared for frozen sections and sliced at 12 µm and subjected to X-gal staining for 1 hr to overnight, according to the signal intensity. X-gal staining was performed as described previously (*Zhu et al., 1998*, *2001*). Co-localization of β-gal expressing cells with other lineage markers was obtained by double immuno-fluorescence labeling with anti-β-gal antibody.

## Golgi staining

Brains from P8 mice were dissected and subjected to Golgi staining for 10 days using a previously published protocol (*Glaser and Van der Loos, 1981*). Brains were then sagittally sectioned at 50 µm on a vibratome and mounted on slides (Leica, Germany). Sections were then developed in the darkroom (20 min 20% Ammonium hydroxide, Sigma-Aldrich + 10 min 20% Kodak fixer, Kodak, Rochester, NY) and dehydrated in ethanol and xylene.

## BrdU assay

Neonatal mice received a single injection of 50 µg/g (gram, body weight) BrdU (Sigma-Aldrich). For proliferation assay, mice were perfused with 4% PFA 2 hr after the last pulse. For cell cycle and differentiation assay, neonatal mice were injected with BrdU 24, 32 and 48 hr prior to analysis at P8. For P8–P18 BrdU differentiation assay, mice were sacrificed 10 days after the initial pulse. In both assays, brains were dissected and processed for paraffin-embedded sections. BrdU immunofluorescence was performed as described previously (*Wang et al., 2009*).

## Neuron/glia co-culture, granule cell trans-well migration assay, and in vitro granule cell differentiation assay

Cerebella from P6 control and *Nf1*[Math1]CKO were dissected and dissociated into single-cell suspension using a previously published protocol (*Lee et al., 2009*). Granule cells and glia were separated and cultured differently. For neuron/glia co-cultures, cultured glial cells were released by trypsination. Following trypsin inhibition, glial cells were counted, spun down, re-suspended in culture medium, and added to 1-day-old cultures of purified cerebellar neurons at a ratio of 2:1. Trans-well migration assay was performed as described previously (*Watkins and Sontheimer, 2011*) with the following modifications. Millicell cell culture inserts (Millipore) were placed into 24-well culture plate (Corning Inc., Corning, NY) and granule cells ($5.0 \times 10^3$) were added in each upper inserts. The bottom wells contained 10 µg/ml fibronectin as chemo-attractant. They were then incubated for 4 hr in a 37°C humidified $CO_2$ incubator. At the end of the experiment, remaining cells in the well (top) were stripped with Q-tips. Cells on the opposite side (bottom) of the insert membrane were fixed with 4% PFA for 30 min and stained with DAPI. The number of cells migrated through the membrane was counted based on high-power images of membranes and the average cell number of 5 areas (up, down left, right, center) and was subjected to statistical analysis. For in vitro differentiation assay, dissociated cerebellar granule cell progenitors were plated on Poly-D-Lysine-coated cell culture plates in growth media without EGF and FGF. Cells were fixed after 72 hr and stained for MAP2 and DAPI. Both trans-well migration assay and in vitro differentiation assay were repeated with cells obtained from at least three mice.

## Western blot analysis

Snap-frozen tissues were homogenized in lysis buffer for 20 min on ice and subjected to centrifugation at 14,000 rpm for 10 min at 4°C. Equal amounts of protein samples were mixed with 1X SDS loading buffer [50 mM Tris-HCL (pH6.8), 2% SDS, 0.05% bromophenol blue, 10% glycerol, 100 mM β-mercaptoethanol] (Biorad, Hercules, CA). Samples were separated on SDS-PAGE gel and transferred onto NC membranes (Millipore). The blots were then blocked in 5% non-fat milk in TBST, followed by incubation of primary antibodies at 4°C overnight. After washing, the blots were incubated

in horseradish peroxidase (HRP)-conjugated secondary antibodies at room temperature for 1 hr. Signals were detected using ECL or ECL plus (GE healthcare, UK) followed by film development. The primary antibodies used were as follows: p-Erk, Erk, p-S6, S6, p-Akt, Akt (1:1000, rabbit, Cell Signaling) and Nf1 (1:1000, rabbit, Upstate, Lake Placid, NY).

## Rapamycin treatment

Rapamycin (Millipore) was dissolved in 100% sterile-filtered ethanol at a concentration of 10 mg/ml and resuspended in vehicle (5.2% PEG400 with 5.2% Tween80, Sigma-Aldrich) at a final concentration of 0.4 mg/ml. The solution was administered by intraperitoneal injection at the dosage of 4 mg/kg (body weight) every other day from P0.5–P21. Rapamycin-treated mice were collected and compared to littermate control and *Nf1*<sup>hGFAP</sup>CKO mice treated with vehicle.

## Rotarod test

Motor function and motor learning were evaluated by rotarod apparatus. 1 hr before the rotarod test, mice were transferred from the holding room to the experimental room. Mice were tested for 4 consecutive days at P21–P24 and P45–P49. The speed of the rotarod was constantly accelerated from 4 to 34 rpm over a 5-min period. The latency to fall from the rod was recorded. The test was stopped at 12 min.

## Quantification and statistical analysis

Anatomically comparable sections from control and mutant brains were visualized under an Olympus BX51 microscope. Images were captured and subjected to analysis. Lengths, areas and the number of cells were quantified using the ImageJ software package. Statistical analysis was carried out using unpaired two-tailed Student's t-test. Comparisons across more than two groups were based on Anova test. At least three animals from each group were used for quantification. Data were presented as mean ± Standard Error Mean (SEM). $p < 0.05$ was considered to be statistically significant.

When quantifying cells in specific folia, we used the following criteria to define the areas to quantify. 'Folia V/IV' refers to the junction between folia V and IV at 20× magnification (i.e., *Figure 1A* middle panels) and the quantification covered this entire area. Similarly, 'Folia IX/X' refers to the junction between folia IX and X at 20× magnification (i.e., *Figure 5—figure supplement 2A*). 'Folium V' or 'Folium X' refers to 40× images in these folia close to the center of the cerebellum that were imaged at comparable positions in controls and mutants.

## Acknowledgements

We thank Dr L Parada for support during the early phase of the study; G Tomasek for technical assistance; members of the Zhu lab for support; Drs A Messing for providing hGFAP-cre mice, D Rowitch for Math1-cre mice, L Chang for Nestin-creER mice, N Heintz for BLBP antibody, B Novitch for Olig2 antibody; Drs R Packer, C Wang, and J Guan for critically reading the manuscript. This work is supported by grants from the DOD (W81XWH-11-1-0251), NIH/NINDS (1R01 NS053900 and 1R01 NS073762) and NIH/NCI (1U54 CA168512-01) (YZ).

## Additional information

### Funding

| Funder | Grant reference number | Author |
| --- | --- | --- |
| U.S. Department of Defense | W81XWH-11-1-0251 | Yuan Zhu |
| National Institutes of Health | 1R01 NS053900, 1R01 NS073762 | Yuan Zhu |
| National Cancer Institute | 1U54 CA168512-01 | Yuan Zhu |

The funders had no role in study design, data collection and interpretation, or the decision to submit the work for publication.

### Author contributions

EK, YW, Conception and design, Acquisition of data, Analysis and interpretation of data, Drafting or revising the article; S-JK, GGM, Acquisition of data, Analysis and interpretation of data; MB, Acquisition of data, Analysis and interpretation of data, Drafting or revising the article; ESJ, Acquisition

of data, Drafting or revising the article, Contributed unpublished essential data or reagents; TEA, Conception and design, Analysis and interpretation of data, Contributed unpublished essential data or reagents; CW, YEL, J-LG, Acquisition of data, Analysis and interpretation of data, Contributed unpublished essential data or reagents; YZ, Conception and design, Acquisition of data, Analysis and interpretation of data, Drafting or revising the article, Contributed unpublished essential data or reagents

## Ethics

Animal experimentation: All mice in this study were cared for according to the guidelines that were approved by the Animal Care and Use Committees of the University of Michigan at Ann Arbor, MI and the Children's National Medical Center in Washington, DC. The approved protocol number at the University of Michigan is PRO00004481. The project (ID# 30001543) was approved by the IACUC (Protocol#: 309-13-09) of the Children's National Medical Center at Washington.

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
