## [Decision Letter]

Thank you for sending your work entitled “Transient MEK inhibition prevents cerebellar developmental defects and improves long-term motor functions in murine NF1” for consideration at *eLife*. Your article has been favorably evaluated by a Senior editor, a Reviewing editor, and 2 reviewers.

The Reviewing editor and the reviewers discussed their comments before we reached this decision, and the Reviewing editor has assembled the following comments to help you prepare a revised submission.

In this study, the authors conducted genetic analyses to identify the cell types and molecular events that underlie a subset of behavioral deficits associated with Neurofibromatosis type 1. Significantly, the authors show that transient MEK inhibition during a neonatal window prevents cerebellar developmental defects and improves long-term motor performance of Nf1- deficient mice. While the reviewers find the study to be overall nicely conducted and potentially important for the development of Nf1 therapeutics they also identified several points that should be addressed prior to publication. In particular, mechanistic insights into the differential response of the Nf1-GFAP animals to MEK inhibitor (responders vs. non-responders) should be provided. Otherwise, this data is better removed from the study. In addition, the authors should discuss the specificity of the Cre drivers used in the study and its potential effect on the observed phenotypes, as well as improve the presentation of figures to non-experts.

Reviewer #1 comments:

1) Define the genotype of Nf1 CKO mice: NF1flox/− and/or NF1flox/flox? Does Nf1 hemizygosity modify the phenotype?

2) Figures 1 and 3: Provide higher magnification pictures for the double stainings (yellow cells are not very evident in the present pictures).

3) Figure 7: It could be interesting to see if there is a significant difference in rotarod curves between G-responders and P-responders. This could strengthen the observation of long term rescue of motor function in G-responders.

4) Discussion: elaborate some hypothesis to explain the difference in response to MEK inhibition of mutant mice based on the required revisions above. Genetic heterogeneity? Were the mice in a controlled genetic background? Drug distribution/bioavailability? (PK data are not provided and they are needed); Weak or inconsistent modulation of the target? (Phospho-Erk expression analysis of tissues from treated mice vs. untreated by WB and/or IHC is not shown or mentioned).

Reviewer #2 comments:

There are several points that should be addressed prior to publication of this study: 1) Although the authors have carefully restricted their analysis of the NF1 deletion studies they have performed to the development of the cerebellum, two of the three Cre driver lines they have employed cause recombination in other regions of the developing brain. For example, the hGFAP-Cre driver line generates strong recombination astrocytes in many brain structures, including the hippocampus, and corpus callosum. Math-1 mediated Cre recombination has not been extensively characterized, but Math1 is known to be required for development of a range of brainstem neurons involved in proprioception, interoception, hearing and arounsal Rose et al, 2009). The authors have made little effort to make the reader aware of these facts, and how they may affect the interpretation of their data. While this point is unlikely to be important for evaluation of anatomical defects in cerebellar development, the behavioral data they have provided must be considered in the light of this knowledge. The authors should substantively revise the manuscript to educate the reader with regard to these points, the potential impact they have on our interpretation of this work, and their considerations regarding the potential therapeutic effects of MEKi.

2) It is unclear to this reviewer that the observation of Tbr+ cells in the EGL and ML of Nf1/Math1 CKO mice proves that loss of NF1 causes altered fate-specification. Of course, these cells might represent GCs that are mis-specified, as the authors have suggested. Alternatively, they could represent ectopic UBCs that were correctly specified in the rhombic lip, but followed an abnormal migratory path to arrive in the EGL and ML. The fact that they are labeled in the EGL by BrDU does not distinguish between these two possibilities. In my opinion, much more precise and detailed characterization of these ectopic cells would have to be done to conclude that they are mis-specified.

3) The figures presented are exceedingly complex! Although the data are very nicely presented, I believe the density of the data presented in these figures will cause all but the most dedicated scientists to disengage from the study before they have finished reading it. Although I would not reject the study on such superficial grounds, a careful culling of the data so that only the essential points are presented in the primary figures would significantly improve the paper.

---

## [Author Response]

*In this study, the authors conducted genetic analyses to identify the cell types and molecular events that underlie a subset of behavioral deficits associated with Neurofibromatosis type 1. Significantly, the authors show that transient MEK inhibition during a neonatal window prevents cerebellar developmental defects and improves long-term motor performance of Nf1- deficient mice. While the reviewers find the study to be overall nicely conducted and potentially important for the development of Nf1 therapeutics they also identified several points that should be addressed prior to publication. In particular, mechanistic insights into the differential response of the Nf1-GFAP animals to MEK inhibitor (responders vs. non-responders) should be provided. Otherwise, this data is better removed from the study*.

We believe that these results are important parts of the study and should be included in the manuscript. Accordingly, we have included additional data to support our conclusion that the extremely high levels of Erk activation in *Nf1*-deficient glial cells (compared to other cell types) in the developing cerebellum is the major mechanism underlying differential responses in the cerebella of MEKi-treated *Nf1*^hGFAP^CKO mice (Good-responders or G-responders versus Poor-responders or P-responders). Furthermore, we have included a new section subtitled by “Clinical Implication” in the Discussion to address this important issue.

It should be emphasized that we employed a “MEKi in milk” strategy to deliver MEKi to neonatal pups through milk from lactating females treated with MEKi at a dose of 5 mg/kg body weight. While this delivery strategy causes minimal physical distress to pups, it also leads to inconsistent uptake of MEKi from milk due to physical weakness of some of *Nf1*^hGFAP^CKO pups. Western blot analysis showed that this “MEKi in milk” strategy achieved a mild 30% to 50% of Erk inhibition in the developing cerebella of MEKi-treated *Nf1*^hGFAP^CKO pups.

Strikingly, mild Erk inhibition with this protocol robustly and consistently rescued developmental defects in *Nf1*^Math1^CKO cerebella, and over 60% of the *Nf1*^hGFAP^CKO cerebella (G-responders). It should be noted that the remaining P-responders are not “non-responders”, as these MEKi-treated *Nf1*^hGFAP^CKO mice still exhibited a complete rescue of ectopic Tbr2 expression in their cerebella, as well as consistent rescues in the defects in the corpus callosum and SVZ, as described previously (63). The only observed variation is the inconsistent rescue in the glial defects observed in the cerebella of MEKi-treated *Nf1*^hGFAP^CKO mice, which is likely due to the extremely high levels of Erk activation in *Nf1*-deficient glial cells in the developing cerebellum compared to other cells. Indeed, when the dose of MEKi was increased to 20 mg/kg to lactating females, a more consistent rescue in cerebellar defects was observed in the cerebella of *Nf1*^hGFAP^CKO mice. These new data have been included in the revised Figure 7; Figure 7—figure supplement 1 and Figure 7—figure supplement 2.

*In addition, the authors should discuss the specificity of the Cre drivers used in the study and its potential effect on the observed phenotypes*.

We have addressed these points in the “Responses to Reviewer 2’s comments #1” (see below).

*The authors should improve the presentation of figures to non-experts*.

We have provided detailed information on the revision of 6 out of 7 figures in the initial submission (see “Reponses to Review 2 comments #3”).

Reviewer #1 comments:

1) Define the genotype of Nf1 CKO mice: NF1flox/− and/or NF1flox/flox? Does Nf1 hemizygosity modify the phenotype?

All the Nf1 conditional knockout (CKO) mice analyzed in this study have the genotypes of Cre+;Nf1^flox/flox^. This information was included in the “Materials and Methods” section subtitled by “Control and Mutant Mice“. It should be noted that no significant phenotypic difference was observed between hGFAP-cre+;Nf1^flox/flox^ and hGFAP-cre+;Nf1^flox/-^ mice, as previously described (Zhu et al. Development 2005).

*2)*
Figures 1 and 3*: Provide higher magnification pictures for the double stainings (yellow cells are not very evident in the present pictures)*.

We have included higher magnification images for the double staining of NeuN and GABAα6 in the insets of the revised Figure 1. Of note, GABAα6 staining is mostly cytoplasmic and surrounds the nuclear NeuN staining. Higher magnification images for the double staining of BLBP/βgal and NeuN/βgal have also been included as insets in the revised Figure 3 (which corresponds to the old Figure 3).

*3)*
Figure 7*: It could be interesting to see if there is a significant difference in rotarod curves between G-responders and P-responders. This could strengthen the observation of long term rescue of motor function in G-responders*.

We did observe that G-responders stayed significantly longer on Rotarod than P-responders, providing a nice correlation between recues in cerebellar developmental defects and motor functions of MEKi-treated *Nf1*^hGFAP^CKO mice. This result has been included in the revised Figure 8.

*4) Discussion: elaborate some hypothesis to explain the difference in response to MEK inhibition of mutant mice based on the required revisions above. Genetic heterogeneity? Were the mice in a controlled genetic background? Drug distribution/bioavailability? (PK data are not provided and they are needed); Weak or inconsistent modulation of the target? (Phospho-Erk expression analysis of tissues from treated mice vs. untreated by WB and/or IHC is not shown or mentioned)*.

The detailed discussion about these points has been included in “Responses to General Comments #1”. In brief, the control and mutant mice analyzed in this study were maintained in the mixed background of C57Bl, 129 Svj and FVB backgrounds. To minimize the effect of genetic background, we analyzed control and mutant mice among littermates. As discussed above, control and mutant littermates used in this study received MEKi through milk from the same lactating females. Based on Western blot analysis, a mild 30% to 50% of Erk inhibition was observed among MEKi-treated *Nf1*^hGFAP^CKO mice. This mild inhibition consistently rescued cell-autonomous defects in neurons in the cerebella of MEKi-treated *Nf1*^Math1^CKO mice, but exhibited differential responses in glial-dependent cerebellar defects. The inconsistent rescue in the glial defects observed in the cerebella of MEKi-treated *Nf1*^hGFAP^CKO mice is likely due to the extremely high levels of Erk activation in *Nf1*-deficient glial cells in the developing cerebellum compared to other cells. Indeed, when the dose of MEKi was increased to 20 mg/kg, a more consistent rescue in cerebellar defects was observed in the cerebella of *Nf1*^hGFAP^CKO mice.

Reviewer #2 comments:

*There are several points that should be addressed prior to publication of this study: 1) Although the authors have carefully restricted their analysis of the NF1 deletion studies they have performed to the development of the cerebellum, two of the three Cre driver lines they have employed cause recombination in other regions of the developing brain. For example, the hGFAP-Cre driver line generates strong recombination astrocytes in many brain structures, including the hippocampus, and corpus callosum. Math-1 mediated Cre recombination has not been extensively characterized, but Math1 is known to be required for development of a range of brainstem neurons involved in proprioception, interoception, hearing and arounsal Rose et al, 2009). The authors have made little effort to make the reader aware of these facts, and how they may affect the interpretation of their data. While this point is unlikely to be important for evaluation of anatomical defects in cerebellar development, the behavioral data they have provided must be considered in the light of this knowledge. The authors should substantively revise the manuscript to educate the reader with regard to these points, the potential impact they have on our interpretation of this work, and their considerations regarding the potential therapeutic effects of MEKi*.

We agree with this reviewer that the Cre drivers except for L7-cre used in this study also target cells in the other regions of the brain (which has been included in the legends of Figure 1—figure supplement 1). Therefore, behavioral data are a collective reflection of rescues of multiple NF1-associated brain structural defects including those in the cerebellum and corpus callosum. Indeed, this is one of the major strengths of this transient MEKi treatment protocol that could potentially exert therapeutic benefits on multiple NF1-associated brain structural defects including both defects in the cerebellum and other brain regions (which are targeted by hGFAP-cre, as this reviewer pointed out). This discussion has been included in the last paragraph of the Discussion section.

Detailed discussion about the potential therapeutic effects of MEKi has been included in “Response to General Comments #1” and “Response to Reviewer 1 Comments #4”.

*2) It is unclear to this reviewer that the observation of Tbr+ cells in the EGL and ML of Nf1/Math1 CKO mice proves that loss of NF1 causes altered fate-specification. Of course, these cells might represent GCs that are mis-specified, as the authors have suggested. Alternatively, they could represent ectopic UBCs that were correctly specified in the rhombic lip, but followed an abnormal migratory path to arrive in the EGL and ML. The fact that they are labeled in the EGL by BrDU does not distinguish between these two possibilities. In my opinion, much more precise and detailed characterization of these ectopic cells would have to be done to conclude that they are mis-specified*.

We agree with this reviewer that ectopic presence of Tbr2^+^ cells in the EGL and ML of the *Nf1*^Math1^CKO cerebellum could be caused by abnormal migration of Tbr2^+^ cells from the RL. However, we regret that we did not make it clear in the initial submission that ectopic presence of Tbr2^+^ cells in the EGL and ML of the *Nf1*^Math1^CKO cerebellum are distinct from those relatively well characterized RL-derived Tbr2^high^NeuN^-^ UBCs in the literature. These two Tbr2^+^ UBC sublineages can be distinguished at least in six aspects: (1) the marker expression, Tbr2^+^NeuN^+^ vs. Tbr2^high^NeuN^-^, (2) the low vs. high percentage expressing CR^high^, (3) the location, ML and IGL vs. IGL, (4) the developmental origin, precursors in the EGL vs. RL, (5) the timing of birth, postnatal vs. embryonic stages, and (6) the migratory pathway, EGL-ML-IGL vs. RL-PWM-IGL (see the revised Results section).

It is still possible that *Nf1* loss causes abnormal migration of the Tbr2^high^NeuN^-^ sublineage from the RL to the EGL, and then switches them to the Tbr2^+^NeuN^+^ UBC sublineage in the EGL. However, no Tbr2^+^ cells were found in the EGL of the mutant cerebellum at E17.5 or P0.5, when Tbr2^high^NeuN^-^ UBCs migrate out of the RL into the PWM (Figure 5—figure supplement 2). This observation argues against the possibility of abnormal migration of Tbr2^high^NeuN^-^ UBCs into the EGL. Indeed, *Nf1* loss has little or no impact on the generation, migration and total number of Tbr2^high^NeuN^-^ UBCs in the mutant cerebellum. Furthermore, in contrast to the precursors of Tbr2^high^NeuN^-^ UBCs that exit cell cycles during perinatal stages, *Nf1*-deficient precursors of Tbr2^+^NeuN^+^ UBCs continue proliferating in the EGL at least until P8 at the peak of GCP proliferation. Together, these observations suggest that Nf1 plays a specific role in the Tbr2^+^NeuN^+^ UBC sublineage by controlling the fate-specification of GCPs in differentiating into GCs versus Tbr2^+^NeuN^+^ UBCs. This discussion has been included in the revised Discussion section.

*3) The figures presented are exceedingly complex! Although the data are very nicely presented, I believe the density of the data presented in these figures will cause all but the most dedicated scientists to disengage from the study before they have finished reading it. Although I would not reject the study on such superficial grounds, a careful culling of the data so that only the essential points are presented in the primary figures would significantly improve the paper*.

We have carefully reviewed the data and revised 6 of 7 figures in the original submission. In the revised manuscript, each figure was simplified and only included the necessary data to support the main conclusion of the particular figure. All the supporting data were moved to figure supplements. Specifically:

Figure 1 only includes the data that describe the neuronal and glial phenotypes of the *Nf1*^hGFAP^CKO cerebellum. The cartoon describing the specificity of different Cre drivers was moved to Figure 1—figure supplement 1.

Figure 2 was reorganized with focus on the glial phenotypes in the developing *Nf1*^hGFAP^CKO cerebellum.

Figure 3 was simplified and focused on the data showing that Nf1 inactivation in glial precursors at early stages caused the abnormalities similar to those seen in the *Nf1*^hGFAP^CKO cerebellum. The remaining supporting data were moved to a figure supplement.

Figure 4 was simplified by moving 48-hour BrdU pulse-chase data and in vitro data to a figure supplement. In addition, the cartoon summarizing the data was included in Figure 4.

Figure 5 was simplified to include the data describing cell-autonomous and non-cell-autonomous defects of Tbr2 UBCs caused by Nf1 inactivation in the adult cerebellum. Figure 5–figure supplement was split into two figures, which describes glial-independent and –dependent UBC defects, respectively.

Figure 6 is not as complicated as other figures and no change was made.

Figure 7 was split into two figures – the revised Figures 7 and 8. The revised Figure 7 only included the data of short-term effects of MEKi treatment and added new data using higher dose of MEKi in Figure 7. The new revised Figure 8 included the long-term effects of MEKi treatment and behavioral tests and two panels to summarize the main findings of the study. One more figure supplement was included to support the revised Figure 7.